# Identification of significant chromatin contacts from HiChIP data by FitHiChIP

Sourya Bhattacharyya[1], Vivek Chandra[1], Pandurangan Vijayanand[1,2,3] & Ferhat Ay [1,3]

HiChIP/PLAC-seq is increasingly becoming popular for profiling 3D chromatin contacts among regulatory elements and for annotating functions of genetic variants. Here we describe FitHiChIP, a computational method for loop calling from HiChIP/PLAC-seq data, which jointly models the non-uniform coverage and genomic distance scaling of contact counts to compute statistical significance estimates. We also develop a technique to filter putative bystander loops that can be explained by stronger adjacent loops. Compared to existing methods, FitHiChIP performs better in recovering contacts reported by Hi-C, promoter capture Hi-C and ChIA-PET experiments and in capturing previously validated promoter-enhancer interactions. FitHiChIP loop calls are reproducible among replicates and are consistent across different experimental settings. Our work also provides a framework for differential HiChIP analysis with an option to utilize ChIP-seq data for further characterizing differential loops. Even though designed for HiChIP, FitHiChIP is also applicable to other conformation capture assays.

[1] Division of Vaccine Discovery, La Jolla Institute for Immunology, 9420 Athena Circle, La Jolla, CA 92037, USA. [2] Respiratory Biomedical Research Unit, University of Southampton, University Road, Southampton SO17 1BJ, UK. [3] School of Medicine, University of California San Diego, 9500 Gilman Drive, La Jolla, CA 92093, USA. Correspondence and requests for materials should be addressed to F.A. (email: ferhatay@lji.org)

Even though the invention of high-throughput chromosome conformation capture (3C) techniques (e.g., Hi-C[1–3], chromatin interaction analysis with paired-end tag (ChIA-PET)[4]) has revolutionized the three-dimensional (3D) genomics field, it remains costly to generate kilobase resolution contact maps that allow for de novo identification of interacting regulatory elements[3]. Two new techniques that combine Hi-C with chromatin immunoprecipitation-sequencing (ChIP-seq), namely HiChIP (Hi-C chromatin immunoprecipitation)[5] and PLAC-seq (proximity ligation assisted ChIP-seq)[6], show significant improvement over ChIA-PET[4] in direct profiling of regulatory (e.g., H3K27ac) and structural (e.g., cohesin) interactions with moderate sequencing depth (~200 M reads) and in primary cells. However, at present, computational identification of a functionally important subset of interactions/loops/contacts from these data remain difficult. The original articles describing both assays[5,6] (we use HiChIP to refer to both hereafter) use Hi-C-specific computational methods (HiCCUPS[3] or FitHiC[7]) for loop calling from HiChIP data. HiCCUPS detects loops using local neighborhoods to compute an enrichment for the center pixel in each region of the contact matrix. FitHiC, on the other end, estimates a background model from the global set of contact counts to find enrichment of each pixel with respect to overall expectation at that genomic distance. Both methods assume that each genomic bin is represented by roughly equal number of overall contacts, an assumption that is not valid for HiChIP and other targeted conformation capture assays such as ChIA-PET and promoter capture Hi-C (PCHiC)[4,8]. Several other computational methods for Hi-C data, which account for zero-inflation and overdispersion of contact counts[9] and for dependency of contacts among adjacent fragment/bin pairs[10], are also not readily applicable to HiChIP data. On the other hand, several tools developed for ChIA-PET analysis do not support finding loops involving non-peak regions[11–13], a task that is important for HiChIP, which has a broader coverage compared to ChIA-PET[14].

HiChIP signal also depends on the density and distance of restriction enzyme (RE) cut sites with respect to nearby ChIP-seq peaks (1D), and more so compared to ChIA-PET[14]. A recent tool for HiChIP data, hichipper[14], provides a correction for this RE site bias by introducing a new background parameter in MACS2[15] to model the distance between peaks and their nearby RE sites. This correction is used for 1D peak calling from HiChIP data, which is followed by loop calling using MANGO[11]. A more recent tool, MAPS,[16] does not explicitly correct for this peak to RE site distance effect, which is also the case for our method. MAPS adopts a zero-truncated Poisson regression model formerly used for Hi-C data[17] to compute normalized HiChIP contact counts, and uses these normalized counts to compute a statistical significance for each observed count. We provide extensive comparisons of our tool to both of these existing methods using a number of different and complementary metrics given the lack of a gold standard validation set.

Here we develop a versatile method, FitHiChIP, which performs loop calling (i.e., identification of significant contacts) from HiChIP data by: (i) Learning the dependency between assay-specific biases or coverage values for each genomic distance using a regression model. (ii) Smoothing the learned parameters across different distances using a monotonically non-increasing smoothing spline fit. (iii) Computing statistical significance using the learned parameters and corresponding expected counts from a background model inferred either from all possible pairs of peak bins (bins that overlap with provided peak annotations), which we name peak-to-peak or stringent (S), or from pairs involving at least one peak bin, which we name peak-to-all or loose (L). (iv) (Optional) Improving the specificity of the resulting

loop calls further by merging adjacent loops identified as connected components of the binary loop call matrix and then filtering bystander loops that can be explained by putative direct loops that are stronger. FitHiChIP workflow is outlined in Fig. 1a and a pictorial description of the merging filter is provided in Supplementary Fig. 1. Other features of FitHiChIP include: (i) allowing users to either infer peaks from the 1D coverage of their HiChIP data or input a predefined reference set of peaks potentially from a matching ChIP-seq experiment, (ii) reporting significance for: (a) only pairs of bins that both overlap provided peaks (peak-to-peak foreground, similar to ChIA-PET pipelines), (b) pairs that have a peak overlap for at least one side (peak-to-all foreground, similar to PCHiC), or (c) all pairs (all-to-all foreground, similar to Hi-C), (iii) allowing the use of normalization/bias factors either computed from a matrix balancing method or simply from marginalized HiChIP coverage values.

When run on multiple published HiChIP datasets, FitHiChIP identifies loops that better recover contacts reported by in situ Hi-C, PCHiC, and ChIA-PET data in matching cell types compared to existing methods. FitHiChIP also captures previously validated enhancer interactions for several genes including *MYC*, *TP53*, and *NMU*. FitHiChIP results are reproducible among biological replicates and consistent across experiments with varying amounts of starting material, hence robust to experimental and technical variation. By simulating HiChIP contact maps from Hi-C data sampled proportional to ChIP-seq coverage of each bin, we show that FitHiChIP is able to recover the stronger Hi-C loops in the underlying data. This recovery is hampered when coverage values are shuffled before simulation, suggesting that FitHiChIP calls are specific. These simulation results also show that nearly one-third of FitHiChIP loops from actual HiChIP data cannot be explained by the combination of Hi-C and ChIP-seq data highlighting the existence of contacts that are specifically enriched by HiChIP. Our differential analysis results show the importance of intersecting the discovered contact count differences with FitHiChIP loop calls. Our breakdown of differential loops with respect to their relationship to changes in ChIP-seq signal enrichment between the compared cell types demonstrates that a small set of loops, which cannot be explained by changes in ChIP-seq coverage have substantial differences in their HiChIP signal. FitHiChIP is also applicable to other types of conformation capture assays as evidenced by our results here for PCHiC and by recent work for HiChIRP[18]. FitHiChIP is available at https://github.com/ay-lab/FitHiChIP.

## Results

**FitHiChIP loop calls for publicly available HiChIP data**. We apply FitHiChIP as outlined in Fig. 1a to analyze published HiChIP datasets[5,19] of four cell types (Supplementary Table 1): GM12878, K562, and naive CD4$^+$ T cells (reference genome hg19); mouse embryonic stem cells (mESs) (reference genome mm9), with two different immunoprecipitation targets (histone modification H3K27ac and cohesin as profiled either by RAD21 or by SMC1A antibodies). For inferring the background model we either use the stringent (S) model (peak-to-peak), which estimates higher background contact probability (Supplementary Fig. 2) and, hence, more conservative significance estimates or the loose (L) model (peak-to-all), which reports a larger number of loop calls (Supplementary Table 2). For each cell type, we use FitHiChIP on individual replicates to measure reproducibility as well as on the combined data after merging all replicates to achieve maximum statistical power. In this work, we use either 2.5 or 5 kb fixed-size genomic windows/bins for analyzing HiChIP data. We choose these two window sizes for compatibility with existing literature on HiChIP and Hi-C data

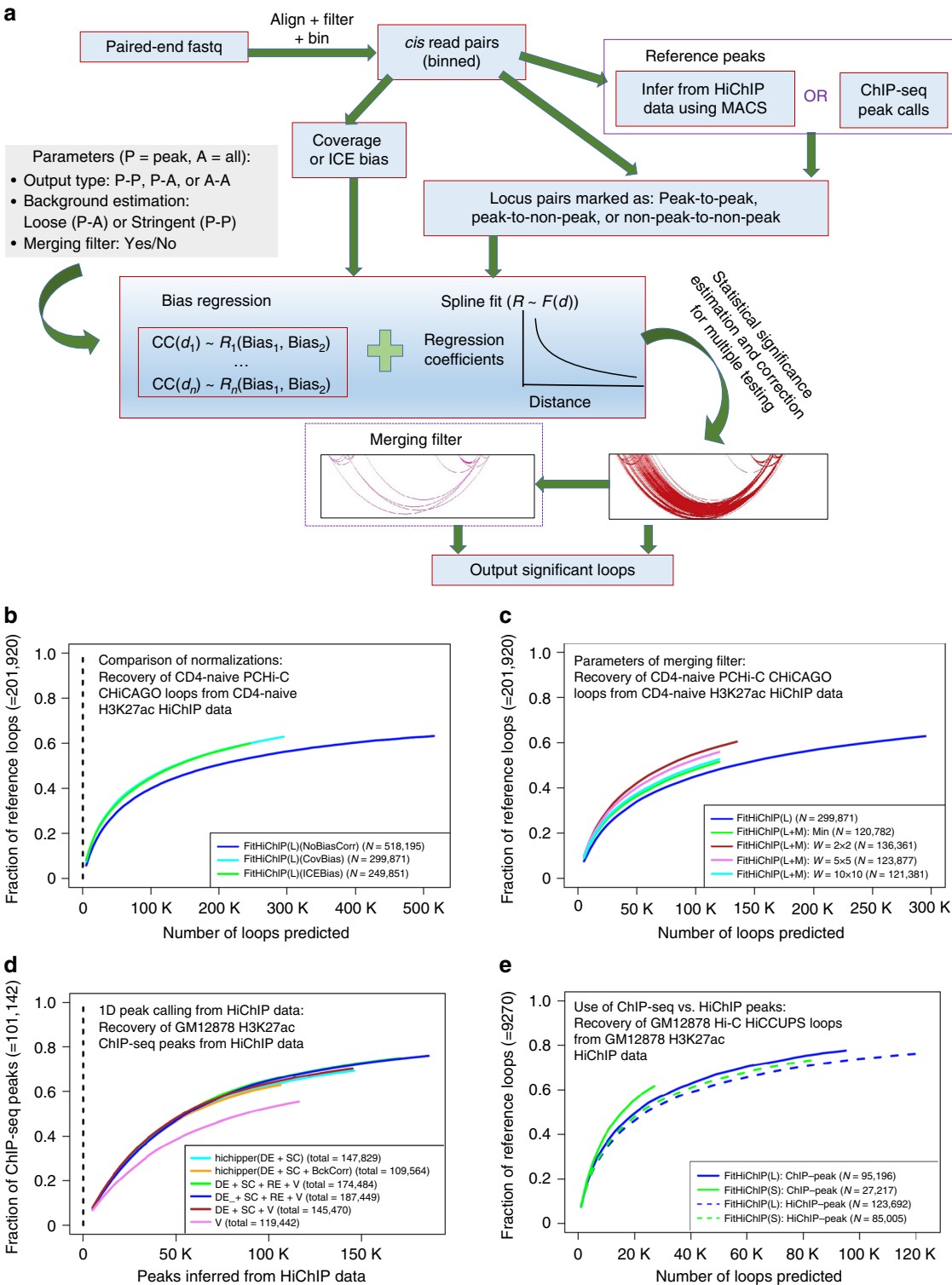

**Fig. 1** Overview and different settings/parameters of FitHiChIP pipeline. **a** Overview of FitHiChIP pipeline. **b** Comparison of different normalization techniques versus no normalization for FitHiChIP in terms of recovering loops from PCHiC data on the same cell type. **c** Recovery performance of PCHiC loops using different settings and window sizes for our merging filter (M) technique. **d** Recovery of ChIP-seq peaks by MACS2 peaks inferred from HiChIP data using different sets of reads. **e** Comparison of the choice of peak calls (either from a reference ChIP-seq data or inferred from HiChIP data directly) in terms of recovering a reference set of loop calls from GM12878 Hi-C data. The symbol *N* indicates total number of loops (or peaks) for the corresponding method. Source data are provided as a Source Data file

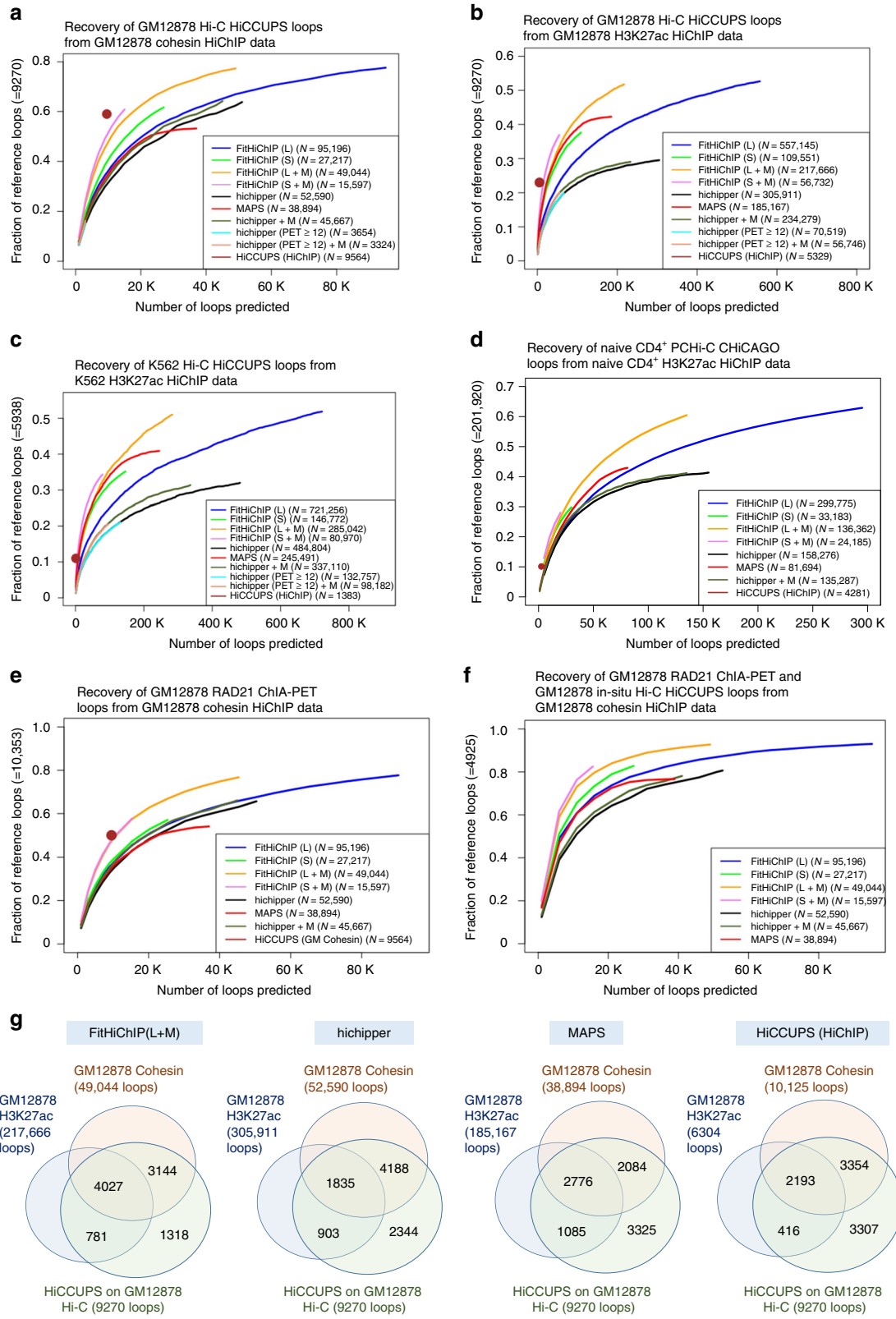

analysis[3,5,14,16,19]. However, users can employ FitHiChIP with any other window size appropriate for their data depending on the sequencing depth and the RE used (Supplementary Note 8). Here, we choose to assign confidence estimates (i.e., use as foreground) the peak-to-all pairs as inclusion of peak-to-non-peak pairs substantially increases the fraction of in situ Hi-C loops that are recovered by FitHiChIP (Supplementary Fig. 3).

**Assessment of loop calls from FitHiChIP and existing methods.** To systematically compare FitHiChIP with existing tools and to evaluate the impact of different parameters, we quantify the extent of concordance with other cell type-matched conformation capture data (Supplementary Tables 3–5). When HiChIP loop calls are compared to a reference set of loops either from Hi-C, PCHiC, or ChIA-PET data, we use recovery plots to measure

**Fig. 2** FitHiChIP recovers a large fraction of Hi-C, PCHiC, and ChIA-PET loop calls. **a–c** Comparative analysis of HiChIP loop calling methods in recovering loops from in situ Hi-C data as called by HiCCUPS. The number on the y-axis represents all HiCCUPS Hi-C loops regardless of their overlap with ChIP-seq peaks. The dark brown dot represents HiCCUPS loops called from HiChIP data with the corresponding number in figure legend representing such loops with at least one end overlapping a reference ChIP-seq peak to make it similar to our peak-to-all foreground. **d**, **e** Similar comparisons with respect to published sets of promoter capture Hi-C and ChIA-PET loops. **f** Similar analysis when the common loops between GM12878 RAD21 ChIA-PET and GM12878 in situ Hi-C HiCCUPS loops are used as the reference set. **g** Detailed analysis of GM12878 Hi-C loop calls from HiCCUPS and the fractions of those calls that overlap with HiChIP loops called on two different datasets with four different methods. For all subfigures, the overlapping loops are determined using a 5 kb slack (see Methods). Source data are provided as a Source Data file

what fraction of reference loops are captured for an increasing number of loop calls (i.e., decreasing stringency threshold) from HiChIP data (Fig. 2). For further comparison with Hi-C data, we create aggregate peak analysis (APA) plots, which measure the enrichment of Hi-C signal for the pair of loci that are deemed interacting from HiChIP data with respect to its local neighborhood (Fig. 3). We also compile a set of long-range validated chromatin loops identified by independent methods (e.g., clustered regularly interspaced short palindromic repeats (CRISPR) screens in single cells and in bulk, DNA fluorescence in situ hybridization (FISH) or 3C) and ask whether these are captured by different methods from HiChIP data (Fig. 4). Furthermore, when biological or technical replicates are available for HiChIP data, we compute and compare the reproducibility of loop calls from different methods as well as the consistency of highly ranked loops across experiments with varying number of cells used as starting material (Fig. 5). Finally, we simulate HiChIP-like contact maps from high-depth GM12878 Hi-C data using ChIP-seq coverages of each bin to test whether FitHiChIP can recover the underlying Hi-C loop calls from simulated data and to see whether this recovery is specific compared to a simulation with shuffled ChIP-seq coverage values (see Methods).

We provide detailed discussions of how the choice of normalization (Fig. 1b and Supplementary Figs. 4–6), the use of a merging-based filtering to eliminate the indirect contacts (Fig. 1c, Methods, and Supplementary Figs. 1 and 7–9) and the choice of using peak calls either from ChIP-seq data or from different read types resulting from HiChIP data after discarding the pairing of reads with or without the correction for RE site distribution (Fig. 1d, e and Supplementary Figs. 10–12) impact FitHiChIP results in Supplementary Notes 1–4.

**FitHiChIP recovers loops from in situ Hi-C experiments**. Here we first compare the performance of FitHiChIP, hichipper, MAPS, and HiCCUPS loop calls from HiChIP data, in terms of recovering high confidence loop calls from in situ Hi-C data of GM12878 and K562 cell lines at 5 kb resolution[3]. For this, we compute the recovery of HiCCUPS loop calls on Hi-C data for FitHiChIP in four different settings (L, L + M, S, S + M) at 1% false discovery rate (FDR), for hichipper in four different settings (≥2 paired end tags (PETs) (default), ≥2 PETs + M, ≥12 PETs, ≥12 PETs + M) at 1% FDR, for MAPS with default settings at 1% FDR, ≥12 PETs and observed over expected count ratio ≥2, and for published HiCCUPS HiChIP loop calls from three different datasets (see Methods). For FitHiChIP, hichipper, and MAPS, the loop calling is carried out for the genomic distance range of 20 kb to 2 Mb. For HiCCUPS HiChIP calls as well as reference datasets used (Hi-C, ChIA-PET, PCHiC), the readily computed calls are filtered to only keep loops within this 20 kb to 2 Mb distance range (Methods). In all cases, FitHiChIP(L + M) provides better overall recovery (maximum value on the y-axis) of Hi-C loops compared to all settings of other tools (Fig. 2a–c). This holds true even for cases when the competing method reports a larger number of loops compared to FitHiChIP (Fig. 2b, c). MAPS performs comparable to some settings of FitHiChIP for the

H3K27ac datasets (Fig. 2b, c), but has lower recovery at any given number of k to pick top-k loop calls in the cohesin data (Fig. 2a). HiCCUPS loops from HiChIP data show comparable or better recovery with respect to all other methods when each method is restricted to have an equal number of loop calls (equal to that of HiCCUPS); however, the overall recovery of HiCCUPS is quite limited (Fig. 2b, c).

In order to further characterize whether the differences in recovery performance of HiChIP loop calling methods are robust, we carry out the same analysis in a number of different settings. One factor that may have a significant impact is the resolution of contact maps as FitHiChIP, MAPS, and HiCCUPS use fixed-sized genomic bins (5 or 10 kb), whereas hichipper works with loop anchors that vary in size (1–70 kb with a median of 2.5 kb)[14]. Our analyses using 2.5 kb resolution contact maps show that FitHiChIP still outperforms hichipper with a substantial margin in both GM12878 and K562 H3K7ac data regardless of whether the original anchor coordinates (raw) are used or they are binned (at 2.5 kb) for hichipper (Supplementary Fig. 13). We then test another potential factor, which is the use of a post-processing step (i.e., merging filter) for FitHiChIP loop calls. Both applying our merging filter to hichipper results and using a more stringent PET threshold of 12 (as suggested by MAPS) do not improve hichipper results substantially (Fig. 2a–c). Finally, instead of loop calls from our application of hichipper and MAPS on the compared datasets, we directly use loop calls for both methods on two GM12878 HiChIP datasets, which are available from the source data files of MAPS[16]. This analysis was intended to see whether the technical differences in application of tools from different groups impact our observations. We observe that, consistent with the results from our own application of MAPS and hichipper, for both cohesin and H3K27ac data, nearly all settings of FitHiChIP perform better than MAPS and hichipper (Supplementary Fig. 14). Overall, these results suggest that the better recovery performance of FitHiChIP cannot be explained by differences in resolution, distance range, post-processing settings, or technical differences in data processing or application of tools.

**FitHiChIP recovers loops from other 3C data**. Next, we utilize loop calls from PCHiC or ChIA-PET as our reference set, instead of Hi-C. FitHiChIP(L + M) again outperforms hichipper and MAPS in overall recovery even for cases when the competing method calls a larger number of loops (Fig. 2d, e and Supplementary Fig. 15). For these reference sets, FitHiChIP with merging filter and HiCCUPS have equal recovery when top-k loops are considered (k equals the number of all loop calls from HiCCUPS) and for GM12878 cohesin data both are substantially better compared to MAPS, hichipper, or FitHiChIP without the merging filter (Fig. 2e). HiCCUPS, however, reports much smaller number of loops, especially for naive CD4+ T cells H3K27ac HiChIP data leading to an overall recovery of only ~10% of PCHiC loops, whereas FitHiChIP recovers over 60% (Fig. 2d). When we use as reference the loops that are consistent in two different types of conformation capture experiments (e.g., Hi-C and ChIA-PET), FitHiChIP still

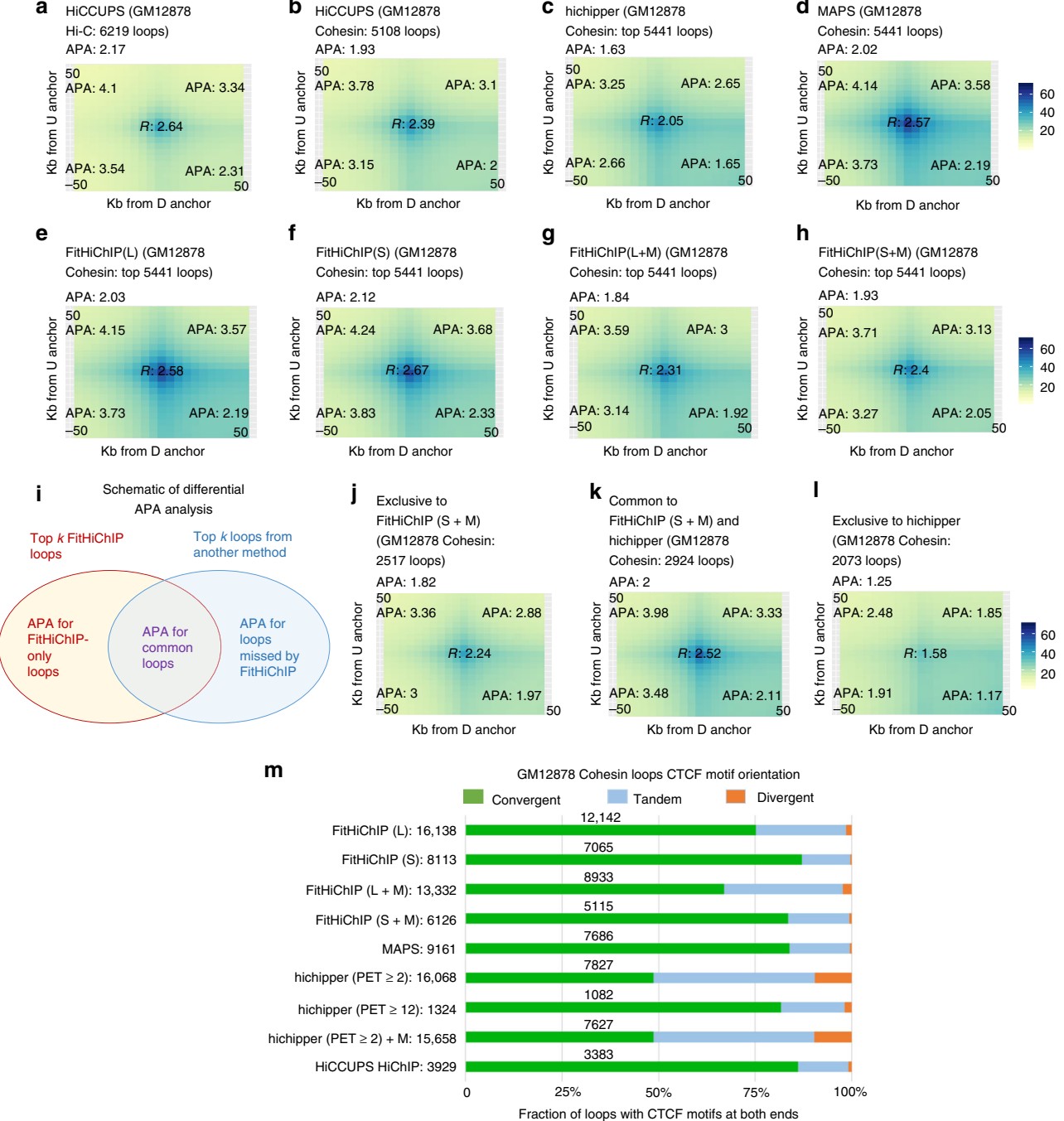

**Fig. 3** FitHiChIP loops are supported by in situ Hi-C data. **a** APA plot for HiCCUPS loops called from GM12878 in situ Hi-C data[3] using the same dataset as the underlying contact map. **b–d** APA scores for HiCCUPS, hichipper, and MAPS loops computed from GM12878 cohesin HiChIP data[5], respectively. **e–h** APA scores for different FitHiChIP versions for the same GM12878 cohesin HiChIP data[5]. For HiCCUPS, all 5108 reported loops are used, whereas for FitHiChIP, hichipper and MAPS, the top $k$ loops are considered, where $k = 5441$ (equal to the number of GM12878 RAD21 ChIA-PET loops). (i) A schematic of the comparative APA analysis for overlapping and exclusive loops between FitHiChIP (top $k$ loops) and a set of loops either from a reference method (containing $k$ loops in total) or from a competing method (top $k$ loops). **j–l** The results of comparative APA analysis for overlapping and exclusive loops between FitHiChIP(S + M) and hichipper for GM12878 cohesin HiChIP data[5]. For all APA plots above, the overlapping loops are determined using 5 kb slack (see Methods) and the loop calls are subsetted to the distance range of 150 kb–1 Mb (as suggested in ref. [11]) for each method before determining the top $k$. **m** The breakdown of HiChIP loops overlapping CTCF binding motifs on both sides with respect to CTCF binding orientation for different HiChIP loop callers. The total number of loops with CTCF motifs on both sides are listed on the left for each method and the number of those with convergent orientation (green) are overlaid on the corresponding portion. Source data are provided as a Source Data file

outperforms existing methods (Fig. 2f and Supplementary Fig. 16a). Due to its stringency, we also utilize HiCCUPS loop calls on HiChIP data as a reference set in order to compare the other three HiChIP loop callers. Recovery plots for loop calls from

HiCCUPS on HiChIP data (Supplementary Fig. 8) or from their intersection with ChIA-PET loop calls in matching datasets (Supplementary Fig. 16b, c) all show that FitHiChIP(L + M) has the best overall performance in each case with MAPS

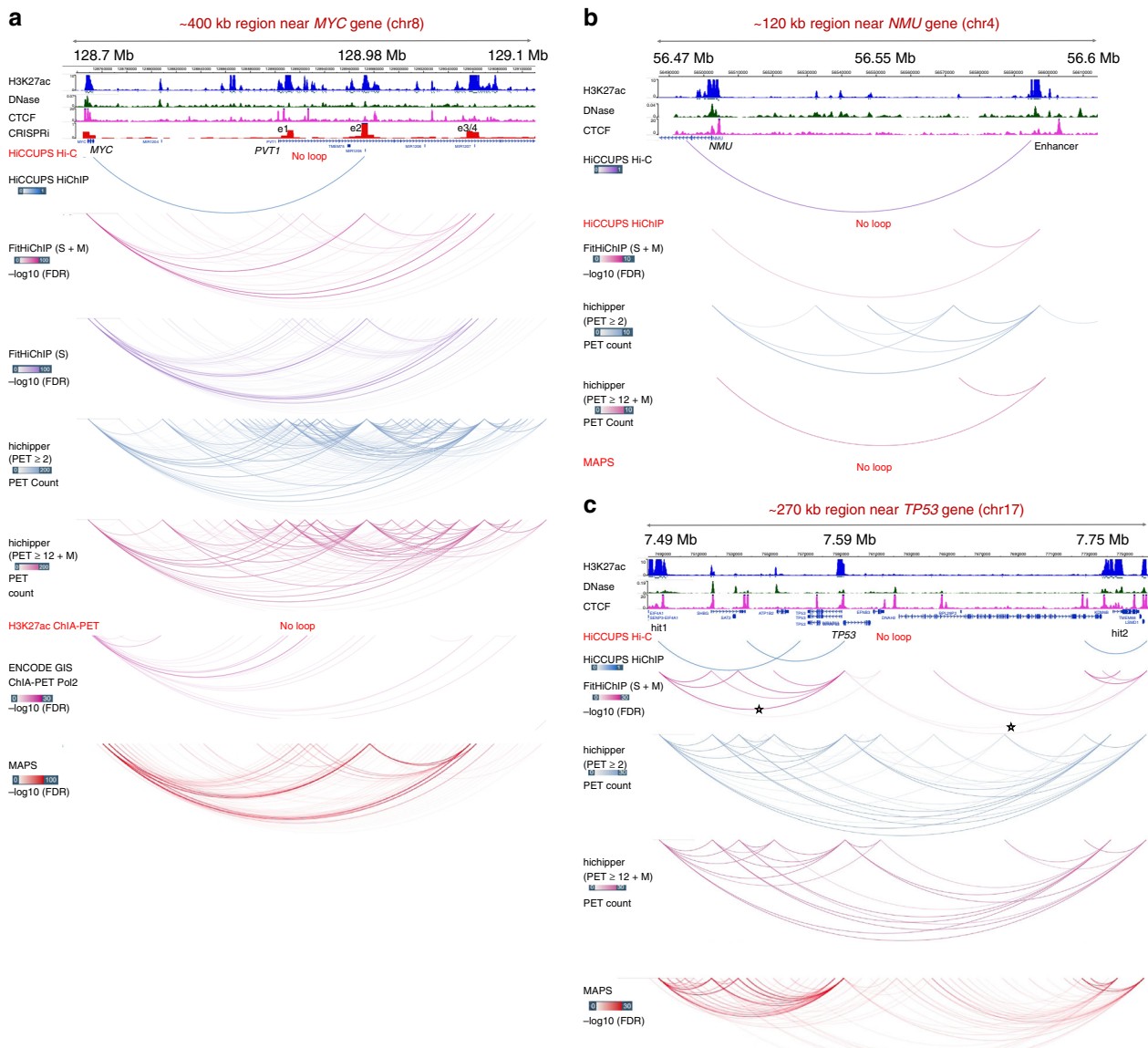

**Fig. 4** FitHiChIP recovers independently validated long-range interactions. **a**: A CRISPRi (clustered regularly interspaced short palindromic repeats interference) screen for *MYC* locus on K562 cells identified seven different enhancer regions[21], four of which are shown here and the remaining three can be seen in Supplementary Fig. 26, which have impact on *MYC* expression when inhibited by a KRAB-dCas9 system (track for CRISPRi score). **b** A single-cell CRISPR screen in K562 cells identified a strong link between expression of *NMU* gene and an enhancer region ~100 kb upstream[24]. **c** Two regions identified by super-enhancer and broad domain analysis coupled with RNA Pol II ChIA-PET data were confirmed to interact with the *TP53* promoter (loops indicated by stars) in K562 cells using EpiSwitch baits[22]. All browser views were generated using WashU Epigenome Browser[46]. For all figures H3K27ac HiChIP data from K562 cells were used for FitHiChIP, hichipper, MAPS, and HiCCUPS HiChIP. Source data are provided as a Source Data file

performing as the second or third best for H3K27ac datasets. We also test the impact of contact map resolution on FitHi-ChIP's performance. Repeating the above recovery analysis using 2.5 kb resolution contact maps, we see that FitHiChIP still has better overall recovery for most settings and better recovery using top-$k$ loop calls for any value of $k$ for all settings compared to hichipper (Supplementary Fig. 17). These results suggest that FitHiChIP's better overall recovery, as well as its recovery when only top-$k$ loop calls are used, is consistent across a wide range of reference data sets.

For GM12878 cell line with cohesin and H3K27ac HiChIP data as well as high-resolution Hi-C loop calls from HiCCUPS, we then compare the consistency among the three datasets for different HiChIP loop callers. Our results show that over 43% of Hi-C loops are captured by FitHiChIP from both cohesin and

H3K27ac data (three-way intersection) compared to <20, 30, and 24% for hichipper, MAPS, and HiCCUPS, respectively (Fig. 2g). Also, less than only 15% of Hi-C loops (1318 out of 9270) cannot be captured using either HiChIP data by FitHiChIP compared to over 25% for all other methods. These results suggest that with appropriate analysis, HiChIP has the power to recapitulate a large fraction of the strongest loops found from Hi-C data with significantly lower sequencing depth, and to detect new loops, which are also supported in Hi-C in the form of enrichment in contact counts as discussed below.

**FitHiChIP loops show enrichment in Hi-C contact maps.** Previous section shows the recovery of loops called from Hi-C and other datasets by FitHiChIP and existing methods. Here we start from HiChIP loops and ask whether they are supported by

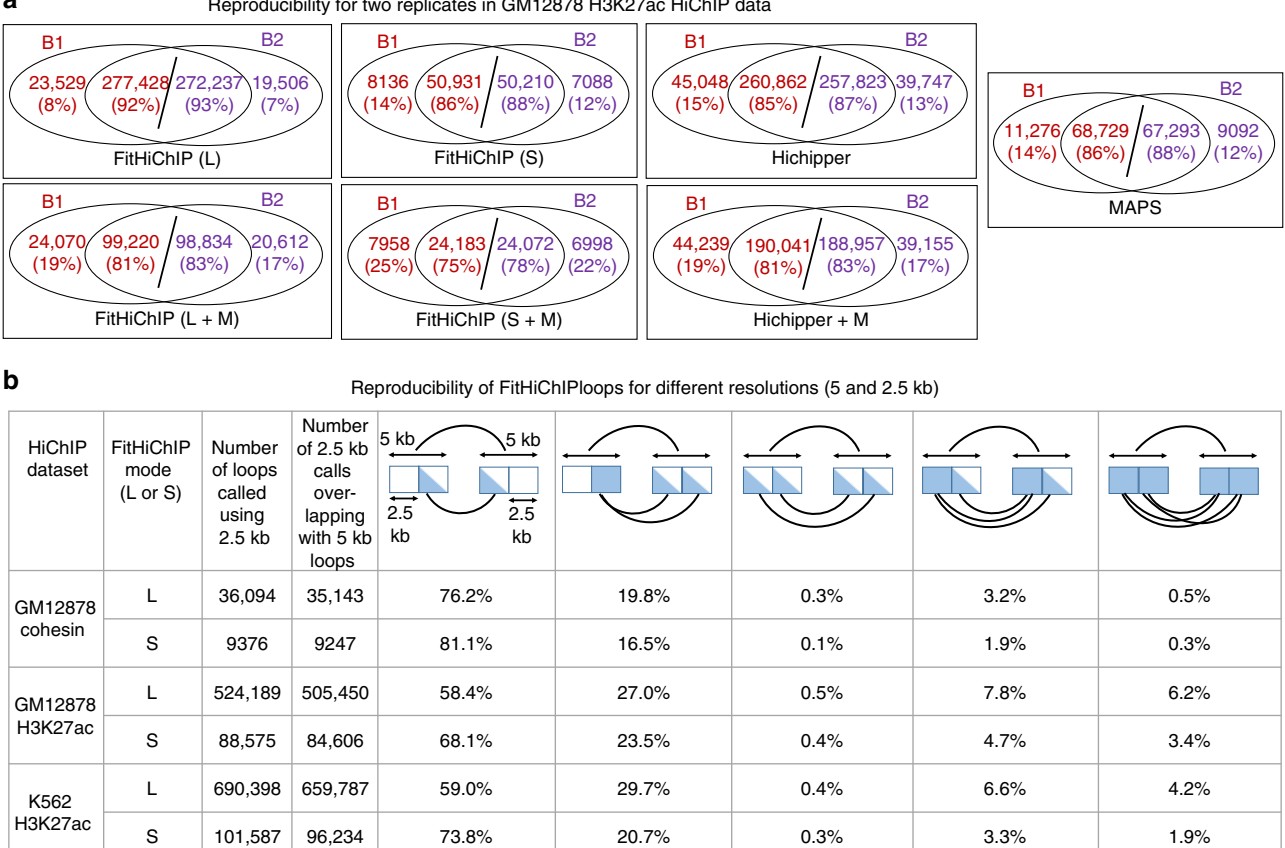

**Fig. 5** FitHiChIP calls are reproducible among replicates and across different resolutions. **a** Overlap between loop calls of different methods for the two biological replicates of GM12878 H3K27ac HiChIP data[19]. Overlapping loops are determined using 5 kb slack (see Methods), and overlap with respect to individual samples are separately shown. **b** Reproducibility of FitHiChIP results between 5 and 2.5 kb resolution loop calls for different HiChIP datasets. The number of all 2.5 kb loops as well as those overlapping with (i.e., contained within) a 5 kb loop call are listed for each dataset. For the 5 kb loops that overlap at least one 2.5 kb loop, the breakdown of five different possible configurations are illustrated with percentages of such cases shown for each dataset. Source data are provided as a Source Data file

Hi-C contact maps. We interrogate: (i) what fraction of loops from HiChIP data are detected by Hi-C loops called using two distinct methods (HiCCUPS[3] and FitHiC[7]), (ii) whether the identified HiChIP loops show an enrichment in Hi-C contact counts when the local contact patterns around them are analyzed in aggregate using APA[11], and (iii) whether the HiChIP loops that are method specific (i.e., reported by FitHiChIP but not by hichipper) show distinct patterns in terms of the support they have from the cell type-matched Hi-C data.

First, when compared to hichipper-specific loops, FitHiChIP loops (either common with hichipper or exclusive) show higher overlap with Hi-C loop calls (Supplementary Fig. 18). Similar comparison between FitHiChIP and MAPS shows that loop calls that are specific to MAPS (in comparison to FitHiChIP) are also well supported by the underlying Hi-C data unlike hichipper (Supplementary Fig. 19). Second, in terms of APA scores (the higher the better), we observe that hichipper loops consistently have the lowest enrichment compared to loop calls from all settings of FitHiChIP and MAPS on both cohesin (1.63–1.69) and H3K27ac (1.27–1.35) HiChIP data[5,19], to RAD21 ChIA-PET loops[20] (1.83), and to HiCCUPS loops from both Hi-C[3] (2.17) and HiChIP data (1.93 cohesin, 1.76 H3K27ac) (Fig. 3 and Supplementary Figs. 20 and 21). In agreement with the previously mentioned overlap analysis, similar to FitHiChIP loops (1.84–2.12 cohesin, 1.49–1.79 H3K27ac), MAPS loop calls are also highly supported by Hi-C data (2.02 cohesin, 1.75 H3K27ac)

(Fig. 3 and Supplementary Figs. 20 and 21). Notably, higher APA scores are not explained by preference of reporting shorter distance loops, because, for both cohesin and H3K27ac datasets, FitHiChIP(S) reports the highest APA scores and has the largest median loop distance. Lastly, when we analyze method-specific HiChIP loop calls (Fig. 3i), loops exclusive to FitHiChIP (for all settings) show higher APA scores than loops exclusive to hichipper (Fig. 3j–l and Supplementary Figs. 22 and 23). A similar comparison with MAPS shows an asymmetry in the method-specific loop counts (higher for FitHiChIP), but a generally comparable APA score suggesting that both FitHiChIP and MAPS-exclusive loops correspond to regions of Hi-C contact enrichment (Supplementary Figs. 24 and 25). A more detailed discussion of all these results is available in Supplementary Note 5.

**FitHiChIP loops highly overlap with convergent CTCF motifs.** As CTCF/cohesin-associated interactions show higher preference in convergent orientation of CTCF motifs[3], we test whether loops called from GM12878 cohesin HiChIP data using different methods also show such preference (Fig. 3m). These results suggest that the percentage of convergent loops is similar among most methods, although the number of reported convergent pairs vary. FitHiChIP(L) identifies the largest number of such loops (over 12 k compared to <3.5 k for HiCCUPS and 7.5 k for MAPS) highlighting FitHiChIP's improved sensitivity in recovering

additional structural loops with expected CTCF-binding configuration compared to other HiChIP methods as well as to loops discovered from GM12878 in situ Hi-C data (3619 convergent pairs)[3].

**FitHiChIP identifies independently validated distal loops.** To assess whether FitHiChIP connects distal enhancers to their experimentally validated target promoters from HiChIP data, we compile a list of loci for which functional data (e.g., CRISPRi) is available together with the HiChIP data for the same cell line[3,21–24]. These include the loci for *MYC*, *NMU*, *TP53*, *MYO1D*, and *SMYD3*[23] genes with different functional experiments aiming to link distal enhancers to their regulation as well as four regions with strong CTCF-dependent long-range loops that are identified from Hi-C and validated by DNA FISH[3]. For the ~400 kb region around *MYC*, both FitHiChIP and MAPS identify all four enhancer regions found from CRISPRi screen as interacting with *MYC* promoter (K526 H3K27ac HiChIP), whereas HiCCUPS[19] only reports one enhancer as interacting and hichipper reports a large number loops in this locus, most of which are short range and not from the *MYC* promoter (Fig. 4a). The loops to the farthest enhancers at ~2 Mb are captured by all loop callers as well as by Hi-C (Supplementary Fig. 26). For *NMU*, FitHiChIP and hichipper accurately capture the enhancer identified from a single-cell CRISPRi screen[24] as interacting with the promoter, whereas both HiCCUPS and MAPS fail to do so (Fig. 4b). In case of *TP53* promoter, FitHiChIP identifies loops to both hit regions identified using EpiSwitch baits[22], whereas HiCCUPS misses both (Fig. 4c). On the other hand, hichipper and MAPS report a large number of loops including the *TP53* promoter with no specificity to the two hit regions. We discuss the results for *MYO1D* and *SMYD3*[23] genes as well as for FISH-validated loops in detail in the supplementary (Supplementary Notes 6 and 7 and Supplementary Figs. 27–29). Overall, these results suggest that FitHiChIP is able to recover functionally validated or experimentally confirmed contacts/interactions from cell type-matched HiChIP data without reporting many potentially false-positive loop calls.

**Reproducibility and robustness of FitHiChIP loops.** We also evaluate to what extent the loop calls of FitHiChIP and other methods from HiChIP data are reproducible across technical and biological replicates (Fig. 5a and Supplementary Figs. 30 and 31). In general, all three HiChIP loop callers have better reproducibility when compared to published loop calls from replicates of other assays such as Hi-C and ChIA-PET (Supplementary Note 8). We also compare FitHiChIP calls from mES HiChIP samples generated from varying number of cells as starting material (cohesin with 1, 5, and 10 M cells, H3K27ac with 50 K, 100 K, 500 K, and 25 M cells)[5,19]. These results also show a significant overlap of FitHiChIP loop calls with samples generated using higher cell numbers and higher sequencing depth, leading to a substantially higher number of loops/discoveries as expected (Supplementary Note 8 and Supplementary Fig. 32). Finally, we compute the overlap of FitHiChIP loop calls from HiChIP data that is binned at 2.5 or 5 kb resolution (Fig. 5b, Supplementary Note 8, and Supplementary Fig. 33). Our results for three different HiChIP datasets show that, both for loose and stringent background models, over 95% of 2.5 kb loops are also detected by an overlapping 5 kb loop (Supplementary Fig. 33). Among the 5 kb loops that overlap at least one 2.5 kb loop, we see that most 5 kb loops can be resolved to a single underlying 2.5 kb loop (Fig. 5b), suggesting that FitHiChIP calls are, to a certain extent, robust to the choice of contact map resolution.

**Robustness of FitHiChIP in simulated HiChIP data.** We also test the robustness of FitHiChIP by first simulating HiChIP maps using Hi-C and ChIP-seq data such that bins with higher (lower) ChIP-seq signal have higher (lower) Hi-C read coverage after the simulation (Methods). We then apply FitHiChIP on this simulated HiChIP maps, in order to see whether FitHiChIP recovers loops generated by the underlying Hi-C data and by the real/non-simulated HiChIP data. For this purpose, we use in situ Hi-C data[3] and matching ChIP-seq data for cohesin and H3K27ac for GM12878 cells (Supplementary Table 1). Application of FitHiChIP shows that loops identified from the simulated HiChIP maps correspond, to a large extent (85% to 99%), to loops reported either by the underlying Hi-C data or the real HiChIP data (Supplementary Note 9 and Supplementary Figs. 34a and 35a). With respect to HiCCUPS loops from Hi-C data, the simulated cohesin and H3K27ac HiChIP data recovers 72% and 61% of such loops, respectively, whereas FitHiChIP calls from simulated maps that use shuffled ChIP-seq coverages are only able to capture 12% and 14% of the same loops while reporting a very small number of overall loops compared to those from simulated maps with no shuffling of the coverage (Methods, Supplementary Figs. 34b and 35b). These results suggest that both the statistical power and the recovery of reference loops for FitHiChIP is specific to the case when real HiChIP data are used or when realistic (i.e., not shuffled) ChIP-seq coverages are used for simulating HiChIP maps from Hi-C data (Supplementary Note 9).

**Using FitHiChIP in other conformation capture experiments.** We also test the applicability of FitHiChIP in other conformation capture assays such as PCHiC, by using GM12878 PCHiC dataset (combined replicates) provided in Mifsud et al.[8]. We have used the capture design file corresponding to the captured restriction fragment as our reference ChIP-seq peak file required by FitHiChIP. Comparison with respect to the loops generated by CHiCAGO[25], a tool designed specifically for analyzing PCHiC data, shows that FitHiChIP recovers nearly 90% CHiCAGO loop calls (Supplementary Note 10 and Supplementary Fig. 36a). FitHiChIP also has a comparable recovery to CHiCAGO when GM12878 RAD21 ChIA-PET loops, FitHiChIP or HiCCUPS loops on H3K27ac HiChIP data, or HiCCUPS loops from GM12878 in situ Hi-C data are used as a reference loop set (Supplementary Note 10 and Supplementary Fig. 36b–f). These results, as well as a recent work that uses FitHiChIP for analyzing RNA-associated chromosome conformation (HiChIRP assay[18]), highlight that FitHiChIP is useful for analyzing other conformation capture assays.

**Differential loop calling for HiChIP data.** Another utility we implement with FitHiChIP is the ability to identify differential HiChIP loops between two conditions with replicate HiChIP experiments. We do this by using edgeR[26,27] for assessing the significance of differential contact counts between the given conditions followed by overlapping the identified differences with FitHiChIP loop calls from each replicate (Supplementary Fig. 37a, b). As HiChIP contact count signal depends on the ChIP enrichment of the underlying loci, we further categorize the resulting differential loops into different groups with respect to changes in ChIP-seq signal between compared conditions for both loci involved in a given loop (i.e., with respect to presence of changes in 1D signal—either 1D differential or 1D invariant). A similar categorization has recently been used by a study interrogating the role of Notch regulation and dependency on the loops between enhancers and promoters[28]. In a comparison between GM12878 and K562 HiChIP data for H3K27ac, we

identify differential contacts between 1D differential as well as 1D invariant loci, suggesting that a subset of differences in contact counts are strictly caused by the underlying 3D conformation, without changes in the activity or chromatin state of the involved loci (Supplementary Fig. 37a, b). We also show that further filtering of such detected 3D differences by restricting them to be overlapping with a FitHiChIP loop call in at least one input sample (a replicate in one category) produces a loop set with significant enrichment differences in APA analysis of the HiChIP data from the compared cell types as well as significant differences in underlying Hi-C contact counts (Supplementary Note 11 and Supplementary Fig. 37c–e).

## Discussion

Here we describe FitHiChIP, an empirical null-based, flexible computational method for statistical significance estimation and loop calling from HiChIP/PLAC-seq data. FitHiChIP jointly models the non-uniform coverage and genomic distance scaling of HiChIP contact counts using a regression model coupled with spline fitting and further filters bystander interactions using an iterative merging filter on each connected component of adjacent loops. FitHiChIP is fast and memory efficient. An important feature of FitHiChIP is that it provides several choices for peak calling, normalization, filtering adjacent loops, background estimation, and pairs of regions to consider for loop calling. Earlier in the text, we have discussed many of these options and provided justification for our default settings; however, we have kept L, L + M, S, and S + M as four different settings throughout the text. When we compare the loop calls from these four settings for GM12878 cohesin and H3K27ac HiChIP data using exact overlaps, we see that a very large fraction of loops from S + M are also reported by the other three settings suggesting FitHiChIP(S+M) as a good surrogate for loops agreed upon by all FitHiChIP configurations (Supplementary Fig. 38). For both cases, a large number of loops that are reported by L and S but not when merging is used confirms the existence of many strong loops involving surrounding regions of actual loop anchor points, which are removed by merging filter. The largest number of loops fall into the category of *identified by loose background but not with stringent*, which highlights the importance of background choice. For instance, if one aims to find loops between boundaries of loop domains[3], which are demarcated by convergent CTCF motifs and have the highest enrichment in contact counts, it may be desirable to use cohesin HiChIP data with a stringent background and preferably with merging filter enabled for FitHiChIP. On the other hand, if the goal is to find enhancer–promoter interactions within domain[3,29,30] that have some contribution to gene expression, then one may choose to analyze H3K27ac HiChIP data using loose background preferably with merging filter to gather a comprehensive set of calls.

We also believe that our work highlights the overlap between Hi-C and HiChIP data and reconciles some of the differences by using multiple computational methods for each data type. For instance, even though we discover many more interactions from HiChIP data compared to HiCCUPS loops from Hi-C, we showed that a very large fraction (nearly 100% for cohesin and more than 60% for H3K27ac) of such HiChIP-specific loops are indeed reported as Hi-C loops when FitHiC, a more lenient method, is used instead of HiCCUPS for the Hi-C data (Supplementary Fig. 18). This suggests that by targeting a specific factor of interest, HiChIP amplifies the loop signal for pairs of regions enriched for that factor, which readily had higher than expected contact counts in the Hi-C contact map. This observation is further confirmed by our results from simulating chromosome 1 HiChIP maps using Hi-C and ChIP-seq data, which

show that 99% (79%) of loops from the simulated cohesin (H3K27ac) map is supported by Hi-C (Supplementary Figs. 34 and 35). Conversely, a large fraction of the strongest (e.g., HiC-CUPS calls) Hi-C loops can be captured by HiChIP data specifically when cohesin complex is targeted (77%, 52% for H3K27ac) (Supplementary Fig. 3).

In terms of differential analysis of HiChIP data, here we present a framework based on edgeR for detecting differences and FitHiChIP to identify which of the identified differences correspond to loops in one cell type or the other. We further segregate the differential loops according to changes in ChIP-seq coverages for the 1D loop anchor regions. These results show that while the bulk of differences in HiChIP data between two distinct cell lines is due to large changes in ChIP-seq signals, there are still hundreds of loops with strong differential contacts, apparent from HiChIP APA plots and supported by changes in Hi-C data, with no underlying ChIP-seq coverage differences. Since our current approach is limited to one-by-one analysis of locus pairs, we, however, cannot rule out potential indirect effects of 1D changes in nearby regions on the differences we observe for such differential loops that are 1D invariant. Future directions for development of differential HiChIP analysis tools may involve modeling contributions of nearby ChIP-seq peaks or loops involving neighboring regions to further stratify various modes of differential looping.

In sum, our work highlights the importance of analyzing HiChIP data with an appropriate method such as FitHiChIP, which goes beyond the strongest loops (e.g., corners of loop domains or TADs) and identifies, exclusively compared to existing methods, a considerable number of Hi-C/ChIA-PET/PCHiC supported loops or functionally validated interactions from the literature. We strongly believe that FitHiChIP is a critical step towards thoroughly exploring the rich data from HiChIP assay as it facilitates the data interpretation and provides a standardized workflow for HiChIP data analysis.

## Methods

**Visualization of loops calls on Epigenome Browser.** For all HiChIP data analyzed in this work, below are the session IDs for individual cell lines with all relevant loop calls (HiChIP, Hi-C, ChIA-PET, PCHiC), which can be loaded in Washington University Epigenome Browser [http://epigenomegateway.wustl.edu/browser/]. After clicking the provided browser link, the user should input one of these session IDs in the text box labeled *Session bundle ID* (bottom left), click *Retrieve session* and then click *Restore* to visualize the tracks.

- GM12878 cohesin loop calls—Session ID: **787c9250-65fa-11e9-9623-5f9c43c4cfff**.
- GM12878 H3K27ac loop calls—Session ID: **b491c3d0-65f7-11e9-b334-5ff263937318**.
- K562 H3K27ac loop calls—Session ID: **019e06b0-65f4-11e9-921c-577d3df57445**.
- Naive CD4+ T cells H3K27ac loop calls—Session ID: **19d21050-65f9-11e9-a173-99425b87a4ba**.
- mESC cohesin loop calls—Session ID: **7a47bff0-65fb-11e9-a36f-5fdb22c1eda3**.
- Differential analysis carried out on two replicates of GM12878 H3K27ac HiChIP and three replicates of K562 H3K27ac HiChIP datasets—Session ID: **27845860-6573-11e9-822e-5db126207a24**.

**HiChIP datasets used from reference studies.** We use published HiChIP datasets (Supplementary Table 1) from four cell types: GM12878, K562 and naive CD4+ T cells (reference genome hg19); mouse embryonic stem cells (reference genome mm9), with two different proteins or histone marks of interest (H3K27ac and cohesin complex as profiled either by RAD21 or SMC1A antibody)[5,19]. For each dataset, we downloaded the *validpairs.txt.gz* files for individual replicates and analyzed the data either per replicate or after merging all replicates into a single file.

**ChIP-seq data.** For each HiChIP dataset, we downloaded the matching (cell type and antibody) ChIP-seq data (peaks and coverage) either from ENCODE[31] or GEO (Supplementary Table 1).

**Estimation of statistical significance by FitHiChIP**. FitHiChIP derives the expected contact probability among any pair of interacting bins by: (1) modeling the decay of contact probability with increasing genomic distance by a monotonic spline fitting technique[7], and (2) performing a regression between the observed contact counts and the bias values of the interacting bins.

*Equal occupancy binning*: The distance decay model aims to estimate the contact probability $p$ between a pair of loci $l_1$ and $l_2$ at a genomic distance $d = d_{l_1 l_2}$ by a function $f(d)$. Suppose that $N$ denotes the number of all possible (interacting or zero count) locus pairs, and $C$ is the total number of contacts between them. We first sort these pairs by increasing genomic distance and then employ an *equal occupancy binning* on the number of overall contacts $C$ (i.e., the number of valid read pairs within the desired range) such that each of the $M$ bins (default $= 200$) would approximately have $\frac{C}{M}$ contacts. For each individual equal occupancy bin indexed by $j$ ($1 \le j \le M$), let $n_j$ be the number of locus pairs belonging to that bin such that $\sum_{j=1}^{M} n_j = N$ and $S_j$ denote the sum of contact counts for these $n_j$ pairs of loci, such that $\sum_{j=1}^{M} S_j = C$. Then, each $S_j \sim \frac{C}{M}$ because of equal occupancy binning with some tie breaks in genomic distance sorting and the average number of contacts per locus pair for bin $j$ will be $\frac{S_j}{n_j}$. We then translate this average into a *prior contact probability*, $p_j$, for each bin such that bin $j$ is $p_j = \frac{S_j / n_j}{C}$. Further, let $D_j$ be the average interaction distance for all $n_j$ possible pairs of loci within the bin $j$. Using the points $(D_j, p_j)$ for $j = 1, \ldots, M$, FitHiChIP fits a univariate spline[7] $f$, such that for a given locus pair $(l_1, l_2)$ with genomic distance $d$, the expected/prior contact probability can be looked up from the spline fit as $p_{l_1 l_2} = f(d = d_{l_1 l_2})$.

*Selection of the background model*: For peak-to-all foreground (loops reported if they have a peak on at least one side; default setting of FitHiChIP), FitHiChIP uses one of two possible sets of locus pairs as background to perform the equal occupancy binning and spline fitting. The first set uses all possible peak-to-all locus pairs (L for loose) within each bin $j$, to define the values $p_j$ and $D_j$. The second set uses only peak-to-peak loops (S for stringent) for each bin $j$ and, hence, provides a more stringent background with higher background probability $p_j$ (Supplementary Fig. 2), leading to more conservative confidence estimates and a lower number of significant loops.

*Statistical significance estimation without bias regression*: If no bias regression is applied, let $p$ be the prior contact probability for a particular locus pair $(l_1, l_2)$ looked up from the spline fit $f$. Then, probability of observing exactly $k$ contacts between this locus pair is computed via binomial distribution as[7]:

$$\mathrm{Prob}(X = k) = \binom{C}{k} p^k (1 - p)^{C-k}. \qquad (1)$$

The $p$ value of observing $k$ number of contacts between $(l_1, l_2)$ is the cumulative probability of observing $k$ or more contacts between them:

$$P(X \ge k) = \sum_{i=k}^{C} \mathrm{Prob}(X = i) \qquad (2)$$

Finally, we correct the resulting $p$ values for multiple testing using Benjamini–Hochberg procedure[32] to compute $q$ values. A locus pair is deemed significantly interacting if it has a $q$ value $\le a$ given FDR threshold such as 0.01 (used in the current study; default in FitHiChIP).

*Statistical significance estimation with bias regression*: In order to correct for coverage differences across different regions of the genome that may relate to technical biases and differences in how these biases may relate to expected number of contacts for different genomic distance regimen, we apply a bias regression on each individual equal occupancy bin $j$ ($1 \le j \le M$) using one of the following ways to compute bias values:

1. *Coverage bias*: Defined for a fixed-size genomic bin $b_j$ (e.g., 5 kb resolution) as the ratio of its HiChIP coverage to the mean coverage of all the bins having non-zero coverage values with the same peak status (bins overlapping ChIP-seq peaks and those that do not are treated separately).
2. *ICE bias*: Computed per bin using a matrix balancing method such as iterative correction (ICE)[33], as re-implemented in HiC-Pro pipeline[34], which treats all genomic bins identically regardless of whether they overlap a 1D peak (i.e., enriched) or not.

For each equal occupancy bin $j$ having $n_j$ locus pairs and average interaction distance $D_j$, we define the following terms:

1. vector of observed contact counts $K^j = \{k_1, k_2, \ldots, k_{n_j}\}$,
2. vector of bias (coverage or ICE) values $B_1^j$ for the first (smaller genomic distance) interacting locus $= \{b_{1,1}, b_{1,2}, \ldots, b_{1,n_j}\}$,
3. vector of bias values $B_2^j$ for the second interacting locus $= \{b_{2,1}, b_{2,2}, \ldots, b_{2,n_j}\}$.

Using the above definitions, FitHiChIP defines the following bias regression model $\mathbf{R}$ for each bin $j$:

$$\log(K^j) = \mathbf{R}(\log(B_1^j), \log(B_2^j)). \qquad (3)$$

We use a linear regression model implemented by the R package MASS, which minimized AIC (Akaike information criterion)[35] among other options. Hence, the

above regression becomes:

$$\log(K^j) = \beta_0^j + \beta_1^j \log g(B_1^j) + \beta_2^j \log(B_2^j), \qquad (4)$$

where $\beta_{0,1,2}^j$ denote the regression coefficients with $\beta_0^j$ corresponding to the intercept term.

After computing above regression for all such equal occupancy bins $j$ ($1 \le j \le M$), the regression coefficients with respect to the average interaction distance values per bin, $D_j$, are fitted a smoothing spline. Similar to the spline fitted to contact probabilities when bias values are not used, these splines $f_{\beta_0}, f_{\beta_1}$, and $f_{\beta_2}$ all show a decreasing trend with increasing genomic distance, thereby eliminating the need for explicitly modeling the change in contact probability with respect to genomic distance (Supplementary Fig. 2).

Using these splines fitted to parameters learned from the regression model, we then compute the expected contact count $c'_{l_1 l_2}$ between a locus pair $(l_1, l_2)$ with genomic distance $d$ and bias values $(b_1, b_2)$ as:

$$\log(c'_{l_1 l_2}) = f_{\beta_0}(d) + f_{\beta_1}(d)\log(b_1) + f_{\beta_2}(d), \log(b_2). \qquad (5)$$

If $C'$ denotes the sum of expected contact counts for all pairs of loci considered, the expected contact probability of $(l_1, l_2)$ becomes $p'_{l_1 l_2} = c'_{l_1 l_2} / C'$. We use this probability $p'$ similar to FitHiC[7] and as described in Eqs. (1) and (2) above, in a binomial distribution to compute statistical significance estimates. In this study, unless otherwise stated, we report peak-to-all interactions within a distance range of 20 kb to 2 Mb and use the bias correction model.

**Merging filter for adjacent loops**. Suppose a significant loop reported by FitHiChIP or another method is represented by an ordered pair of interacting fixed-size (here 5 kb) bins $(x, y)$ where $x < y$. Two loops $(x_1, y_1)$ and $(x_2, y_2)$ are *adjacent* if their constituent bins are either adjacent or equal, that is, $|x_1 - x_2| \le 1$ and $|y_1 - y_2| \le 1$. If we use a 2D contact matrix to represent all possible pairs of bins, and denote a significant loop between two bins $x$ and $y$ as a nonzero entry in location $(x, y)$, the problem of finding a set of mutually adjacent loops reduces to finding non-trivial connected components of a graph using the 8-connectivity rule[36]. We have used Python package *networkx*[37] to find such components/clusters of adjacent statistically significant loops. For each such component, we extract a subset of loops that are likely representatives of direct interactions (with remaining loops as likely bystanders) in order to improve specificity of our loop calls mainly for regions with large number of adjacent loop calls. A trivial approach is to simply report one loop per connected component that has a minimum $p$ value (denoted as *MIN* approach). However, such an approach has the obvious downside of eliminating meaningful interactions when multiple independent and direct loops fall into the same component. Therefore, we employ an *iterative merging* approach to select a subset $S$ from the set of loops $K$ ($|S| < |K|$) within a connected component. In each iteration, we select the current most significant loop $l$ within $K$ (based on the statistical significance value, or contact count, or any other loop scoring method), and include this loop in the set $S$ if and only if $l$ does not belong within $W = B \times B$ (in terms of bins) neighborhood of any loop already in $S$. We use recovery plots to test multiple values for $B$ (2, 5, and 10) and select the number that performs best in terms of specificity (Supplementary Fig. 7). Unless otherwise stated, we use $W = 2 \times 2$ when merging filter is applied to the results of FitHiChIP and that of existing methods.

**Running hichipper (version 0.7.5)**. Base output directories of HiC-pro pipeline, upon excluding the file *rawdata_allValidPairs*, are provided as the input to hichipper. When using hichipper with reference ChIP-seq peaks, we use the following options: *–min-dist 20000 –max-dist 2000000 –skip-background-correction –skip-diffloop –skip-resfrag-pad –skip-qc –make-ucsc*. When we use peak calling by hichipper we set *peaks: EACH,SELF* option in the configuration file, and employ the options *–min-dist 20000 –max-dist 2000000 –skip-diffloop –make-ucsc –keep-temp-files* during execution. Since the output loops of hichipper do not have fixed-size bins, for a fair comparison with our method, we map the midpoint of each interacting bin of hichipper to the overlapping bin (5 or 2.5 kb depending on the bin size considered). For 5 kb bins, because most hichipper loops are between bins of size <5 kb, this process results in duplicate loop calls for which we then eliminate. Note that this conversion reduces the overall number of hichipper calls and given that it suffers from low specificity in capturing reference sets of loops (all of which are also in fixed-size bins), the reduction is likely to help hichipper with specificity issues with no loss in sensitivity. As the default configuration, hichipper only reports loops with a PET count (last column) of at least two[14]. We also use a more stringent filter of at least 12 PET counts for comparison purposes at the request of a reviewer.

**Merging adjacent hichipper loops**. We test the utility of our merging filter to reduce the set of reported loops on the results of hichipper. This corresponding method is denoted by **hichipper + M**. The loops from hichipper are sorted according to decreasing PET count and a window of $2 \times 2$ bins is used similar to that used for FitHiChIP results.

**Running MAPS**. For individual replicates (.fastq.gz reads) of a given cell type, we have executed MAPS with reference ChIP-seq peaks (same as those used for executing FitHiChIP and hichipper) with the following parameters: $bin\_size = 5000$; $fdr = 2$; $filter\_file =$ "None"; $generate\_hic = 0$; $mapq = 30$; $length\_cutoff = 1000$; $threads = 4$; $per\_chr =$ 'True'. In addition, for loop calling, we use the option –BINNING_RANGE 2000000 to call loops up to 2 Mb distance, a threshold used in the current study for all methods. After executing MAPS for individual replicates, we have provided their respective alignment directories to MAPS to generate loops from the combined replicates.

**Using MAPS and hichipper loop calls from MAPS source data**. For both cohesin and H3K27ac HiChIP data from GM12878, we downloaded the loop calls readily made available by MAPS[16] under their source data file (Supplementary Data S1 —ZIP). As these loops were called using a distance threshold of 1 Mb and only for autosomal chromosomes, we filtered FitHiChIP loop calls as well as the reference datasets similarly for comparison.

**Inferring 1D peaks from HiChIP data**. We have tested different combinations of following four sets of reads for 1D peak calling from reads generated by HiChIP: (1) dangling end (DE), (2) self-cycle (SC), (3) re-ligation (RE), and (4) CIS short-range (<1 kb) valid (V) reads (after duplicate removal)[38]. For each set of reads, we use MACS2[15] with the following parameters: -q 0.01 –extsize 147 –nomodel (default in hichipper[14]) to infer corresponding set of peaks.

**Comparing HiChIP 1D peak calls to ChIP-seq peaks**. We evaluate the output peak sets inferred either by different groups of reads by FitHiChIP or by hichipper with or without its specific background correction for restriction sites by computing their overlap with peaks inferred from matching ChIP-seq data. We compute the overlap between peak calls by allowing 1 kb slack (as used in hichipper[14]). We also compute the overlap at the level of 5 kb bins in order to assess the potential impact of different peak calls in labeling 5 kb bins as peak or non-peak bins.

**Overlap between a pair of loops**. Unless otherwise specified, we have used a slack/extension of 5 kb (+ or − one bin on each side) on both loop sets to compute overlap between a pair of loops. We apply this slack after mapping hichipper, and ChIA-PET loops to the 5 kb bin (or 2.5 kb for hichipper loops during the comparison with 2.5 kb FitHiChIP loops); they most overlap on each side as these methods generally report loop calls with lower than 5 kb in size on each end. For HiCCUPS, which reports a mix of 5 and 10 kb resolution loops, we apply the 5 kb slack regardless of the resolution. Note that this gives slight advantage to HiCCUPS in recovery plots since its 10 kb resolution loops will be padded into 20 kb total on each end, whereas all other methods with 5 kb bins will have 15 kb regions on each end for overlap computation. When reporting the percentage of overlap among different sets of loop calls using non-exact overlap (5 kb slack), we report the overlapping and exclusive loops separately with respect to each individual set. For the comparison of 2.5 and 5 kb loop calls from FitHiChIP, we do not use any slack and require that both loop anchors of the 2.5 kb call are strictly contained within the anchors of a 5 kb loop call to deem the two as overlapping.

**Recovery of in situ Hi-C HiCCUPS loops**. HiCCUPS loops for K562 and GM12878 in situ Hi-C data[3] are obtained from Gene Expression Omnibus: GSE63525 (files GSE63525_K562_HiCCUPS_looplist.txt.gz and GSE63525_GM12878_primary + replicate_HiCCUPS_looplist.txt.gz). We retain only the HiCCUPS loops that have a genomic distance between 20 kb and 2 Mb and ask what fraction of them are recovered when an increasing number (decreasing stringency) of loops are predicted by FitHiChIP or other methods. We compute the overlap (successful recovery) with 5 kb slack as described above.

**Recovery of HiChIP HiCCUPS loops**. We obtain the HiCCUPS loops computed on the published HiChIP datasets (Supplementary Table 3)[5,19]. Aside from using HiCCUPS calls on HiChIP data for comparison purposes, due to high specificity of HiCCUPS calls, we also use them as a reference set and compute the recovery of such calls when comparing other methods or experiments to each other as described above. When used as a reference set, we retain only the HiCCUPS HiChIP loops that have a genomic distance between 20 kb and 2 Mb and overlap with a peak bin as assigned by reference ChIP-seq data on at least one side.

**Recovery of ChIA-PET loops**. We obtain the ChIA-PET loops calls from two previous studies (Supplementary Table 4)[20,39]. After binning at 5 kb resolution and removing duplicates, we compute the recovery of ChIA-PET loops with a genomic distance and peak overlap filter as described for HiChIP HiCCUPS loops above.

**Recovery of common loops between HiCCUPS and ChIA-PET**. We obtain the common loops between a reference set of HiCCUPS loops (either HiChIP HiC-CUPS loops provided in refs. [5,19] or in situ Hi-C HiCCUPS loops provided in reg. [3]) and a reference set of ChIA-PET loops[20,39] subject to a slack of 5 kb. The common loops are binned at 5 kb resolution. The recovery analysis for these loops is carried out with similar genomic distance and peak overlap filters as mentioned above.

**Recovery of PCHiC loops**. Similar to other data types described above, we also use PCHiC loop calls to evaluate existing methods. We obtain PCHiC loop calls for naive CD4+ T cells (Supplementary Table 5)[40] that are computed with CHi-CAGO[25]. We keep loops with a CHiCAGO score of ≥5, and within the distance range of 20 kb to 2 Mb. As PCHiC loops involve promoter segments in at least one end, we use only the promoter-specific loops (loops whose at least one end falls within 5 kb of a reference TSS site) of FitHiChIP or hichipper, for computing recovery of reference PCHiC loops.

**Applying FitHiChIP on PCHiC dataset**. To validate the applicability of FitHiChIP on PCHiC data, we have downloaded PCHiC dataset on GM12878 cell line[25] (GEO: GSE81503). The dataset consists of three biological replicates, which have one, three, and two technical replicates, respectively. The.fastq.gz files for these replicates are merged together, and subsequently processed through HiC-Pro pipeline (version 2.9.0)[34], which aligns the reads by Bowtie2[41] (version 2.3.3.1) with respect to reference genome hg19, assigns to the HindIII restriction fragments, filters by their orientation[38], and de-duplicates using Picard[42]. FitHiChIP uses these valid read pairs together with the bait design file of the PCHiC array as peak calls similar to CHiCAGO[25].

For comparison, we download the CHiCAGO significant loops (score ≥5) for this GM12878 PCHiC dataset from the same GEO repository and ask whether the PCHiC loop calls from FitHiChIP or CHiCAGO better recover loops called from GM12878 in situ Hi-C data by HiCCUPS.

**Aggregate peak analysis**. We use Hi-C contact maps (binned at 5 kb) for GM12878 and K562 cell lines[3] that are normalized by ICE[33] to perform APA analyses of loop calls by different methods on HiChIP data or calls from other experiments such as Hi-C, ChIA-PET, and PCHiC. For each called loop, APA extracts the normalized Hi-C contact counts of all locus pairs 50 kb up- and downstream, which corresponds to a matrix of $21 \times 21$ dimensions for 5 kb resolution. It then aggregates these small matrices centered on each individual loop call to generate an aggregate heatmap and to compute several enrichment scores[11]. The APA score displayed on top of each of plot is the ratio of the central pixel and the mean of pixels 15–30 kb downstream of the upstream loci and pixels 15–30 kb upstream of the downstream loci. The symbol R shown at the center of each APA plot is the ratio of the central element to the rest of the elements in the $21 \times 21$ matrix extracted from Hi-C data. The corner-specific APA score displayed at each corner of each APA plot is the ratio of central element to the mean of individual corner regions defined as 10 kb offset from boundary elements in both up- and downstream loci. True looping (highly significant) interactions are expected to have higher contact counts compared to neighboring bins and, hence, higher APA scores indicate that corresponding loops are highly supported by Hi-C data. For visualization purposes, APA considers loops within distance range 150 kb–1 Mb[11].

As the number of FitHiChIP or hichipper loops are substantially higher than reference HiCCUPS or ChIA-PET loops, we use top-$k$ HiChIP loops (determined by higher statistical significance) for APA analysis, where $k$ is the number of loops reported by the more stringent method, which is either HiCCUPS or ChIA-PET. Also, since HiCCUPS loops for several datasets come with a mix of 5 and 10 kb resolution calls, when dealing with 10 kb loops in APA plots, we pick the 5 kb bin on each side that has the smaller coordinate.

**APA scores for overlapping and exclusive loops**. Let $k$ be the number of reference loops (HiCCUPS or ChIA-PET) within the distance range 150 kb–1 Mb. We then select the top-$k$ loops in terms of higher statistical significance from FitHiChIP within the same distance range and compute their overlap with the reference set of loops, by allowing a slack of 5 kb. We then perform the APA analysis for loops that overlap and for those that are exclusive to one method or the other.

**Overlap between HiChIP and Hi-C loop calls**. In order to find what fraction of the loops we identify from HiChIP data by different settings of FitHiChIP or by existing methods, are also identified from Hi-C data, we employ two different significance calling methods to Hi-C data. We use HiCCUPS[3] as a stringent method with high specificity (results downloaded from the datasets[5,19] mentioned in Supplementary Table 3). We also apply FitHiC[7], a more lenient method with higher sensitivity, on the in situ Hi-C datasets of GM12878 or K562 cell lines at 5 kb resolution. We then use these two reference sets of Hi-C loops to compute overlap with loops called from HiChIP data. Loop overlap is computed by allowing 5 kb slack.

**CTCF motif orientation analysis**. To find the CTCF motif orientation of the GM12878 cohesin HiChIP loops generated either by FitHiChIP or the competing methods, we have used the hg19 CTCF peaks provided in ENCODE [encode-project.org/experiments/ENCSR000DZN (file ENCFF710VEH.bed). The routine motifs of Juicer tool[43] [https://github.com/aidenlab/juicer] was applied on the input

set of HiChIP loops. Loops having CTCF motif information (either + or −) in both interacting bins were only considered, from which we computed the frequency and percentage of loops with convergent, divergent, and tandem orientation CTCF motif pairs.

**Simulating HiChIP data from Hi-C and ChIP-seq.** Using coverage values of each 5 kb genomic bin (*bedtools coverage*) from reference ChIP-seq data (Supplementary Table 1), we simulate HiChIP maps by non-uniformly sampling Hi-C contacts such that the resulting row/column sums correspond to the vector of computed ChIP-seq coverage values. Let us denote this vector by $V$ and denote the intra-chromosomal Hi-C contact map for chromosome 1 of GM12878 at 5 kb resolution[3] as a symmetric non-negative matrix $M_0$. The objective is to transform $M_0$ into a matrix $M_t$ whose row and column sums (corresponding to the coverage values of individual bins) emulate the 1D coverage in $V$ after $t$ iterations. We implement the iterative optimization algorithm provided in[44,45]. First we define the following notations:`

1. $M_0[i, j]$ = contact count of the input Hi-C intra-chromosomal matrix, between bins $i$ and $j$.
2. $M_t[i, j]$ = contact count of the output Hi-C intra-chromosomal matrix, between bins $i$ and $j$, at the iteration $t$.
3. $V[i]$ = reference ChIP-seq coverage of $i$th bin.
4. $M_t[i,]$ = row sum for bin $i$ with respect to the matrix $M_t$.
5. $M_t[,j]$ = column sum for bin $j$ with respect to the matrix $M_t$.

The algorithm performs in alternate iterations, row- and column-wise scaling of the input matrix $M$:

- In the row-wise scaling, $M_t[i,] = \frac{M_{t-1}[i,] \times V[i]}{\sum_{\forall j} M_{t-1}[i,]}$.
- In the column-wise scaling, $M_t[,j] = \frac{M_{t-1}[,j] \times V[j]}{\sum_{\forall j} M_{t-1}[,j]}$.

This algorithm has been previously proven to converge to the desired coverage distribution $V$[44,45]. In our implementation, convergence is declared if either the number of iterations $t$ reaches 500 or the sum of difference between matrices at consecutive iterations becomes less than a predefined threshold $\varepsilon$. For both GM12878 cohesin and H3K27ac-simulated HiChIP datasets, we obtain >0.995 correlation between the row (or column) coverage vector of the resulting matrix $M_t$ and the ChIP-seq coverage vector $V$. Finally, entries in $M_t$ are then further scaled to have the sum of contact counts equal to that of the real intra-chromosomal HiChIP contact matrices (for chromosome 1) of GM12878 cohesin or H3K27ac (merged replicates; mentioned in Supplementary Table 1). The scaled contact matrix is then used for loop calling by applying FitHiChIP(L) with peak-to-all foreground and 20 kb to 2 Mb genomic distance range.

In order to achieve randomization in simulated HiChIP matrices, we randomly shuffle the ChIP-seq coverage values in $V$ before the iterative optimization. We perform five different random shuffling of $V$ to generate five simulated maps. While reporting simulation results, we present the average value across these five shuffled maps.

Overlap of the loop calls from simulated (either shuffled or not) HiChIP datasets with respect to other loop calls is computed similar to real HiChIP data. Briefly, loops within a distance range of 20 kb to 2 Mb on each side are considered and the overlap is computed with a 5 kb slack with respect to each individual set involved in the analysis.

**Differential analysis of HiChIP loops.** In this work, two replicates of GM12878 H3K27ac and three replicates of K562 H3K27ac HiChIP data[19] are used to showcase our differential analysis pipeline. First, edgeR[26,27] using the functions *estimateDisp* and *exactTest* with default parameters is applied to the union set of all peak-to-all locus pairs with non-zero contact count in at least one out of the five replicates (20 M pairs). The results from edgeR are further filtered using an FDR of 5% and an absolute fold change >2, in order to get all significant differences. We refer to this set as differential contact enrichments. These differential calls are then further segregated into five different groups with respect to cell type-specific differences in the underlying ChIP-seq signal (ENCODE[31]) for each end. This is achieved by first classifying each 5 kb bin (total 619,150 bins) using the difference between ChIP-seq coverage values of GM12878 and K562 H3K27ac, which are scaled to have an equal overall coverage. This classification involves application of edgeR with default parameters and an FDR of 5% to the scaled coverage values as well as taking the difference between the two signals. As a result, each bin is assigned to either one of the following three categories:

1. HD (high difference): Significant difference (edgeR) of ChIP coverage between two categories.
2. ND (no difference): Non-differential bins with <25% difference of ChIP coverage between GM12878 and K562.
3. LD (low difference): All the remaining bins which, by definition, are non-differential but have ≥25% difference of ChIP coverage.

Using these three bin-level categories, five different locus pair-level categories for differential calls are created as follows: (1) ND–ND, (2) LD–ND, (3) LD–LD, (4) HD–LD/ND, and (5) HD–HD.

To further improve the specificity of differential calls, for each of the above-described categories, only the differential contact enrichments overlapping with statistically significant loops (using FitHiChIP(S) with an FDR of 1%) in at least one replicate of one cell type are extracted, which are referred to as differential loops. This overlap with loop calls enforces higher stringency and, accordingly, greatly reduces the number of reported differences. These differential loops are further filtered for subsets that are exclusive to either GM12878 or K562 (i.e., significant in at least one replicate of one cell type and none of the other).

For comparison of different sets of differential loops with respect to support from HiChIP data, merged HiChIP replicates of GM12878 and K562 H3K27ac HiChIP data are used to create APA plots for differential calls exclusive to (or up in) each cell type. The distribution of differences in underlying ChIP-seq coverage values are plotted and compared against the null hypothesis that the mean absolute difference is <5% using one-sample $t$ test (R function *t.test*) with a $p$ value threshold of $1e^{-6}$, in order to highlight differences across the three groups of differential loops considered. To find out the difference between cell-specific Hi-C contact counts corresponding to differential loop calls, GM12878 (primary + replicate) and K562 (primary) Hi-C datasets are utilized after scaling the two Hi-C matrices to have an equal sum. The log 2 fold change of K562 contact counts divided by that of GM12878 are plotted for each of the three groups and one-sample $t$ tests are conducted to test whether the mean of each distribution is equal to zero ($p$ value threshold of $1e^{-6}$).

**Reporting summary.** Further information on research design is available in the Nature Research Reporting Summary linked to this article.

## Data availability

The publicly available data sets analyzed in this study are summarized in the Supplementary Tables 1–5 with accession IDs and references. The description of source data underlying all main and Supplementary Figs. is provided as a Source Data file. The actual source data, as summarized by this Source Data file, are provided and archived on Zenodo [https://doi.org/10.5281/zenodo.3255048].

## Code availability

Source code for FitHiChIP, along with its README and execution instructions are available at https://github.com/ay-lab/fithichip. We also provide a code capsule for FitHiChIP on Code Ocean [https://codeocean.com/capsule/290e862d-0319-44e9-b192-d13eb394126b].

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

## Acknowledgements

We would like to thank Abhijit Chakraborty, Arya Kaul, and all members of the Ay lab for their input and helpful comments on the tool development. We would also like to thank Halil Tuvan Gezer for creating Singularity and Docker containers of FitHiChIP. This work was funded in part by NIH Grants R35-GM128938 (F.A.) and R24-AI108564 (P.V., F.A., and others).

## Author contributions

S.B. and F.A. conceived the project and designed the method. S.B. implemented the software under supervision of F.A. S.B and F.A. conducted the experiments, interpreted the results, and drafted the manuscript with input from V.C. and P.V. All authors read and approved the final version of the manuscript.

## Additional information

**Competing interests:** The authors declare no competing interests.

