## [Peer Review File · Nature Communications]

Editorial Note: This manuscript has been previously reviewed at another journal that is not operating a transparent peer review scheme. This document only contains reviewer comments and rebuttal letters for versions considered at Nature Communications .

Reviewers' comments:

Reviewer #1 (Remarks to the Author):

The authors describe FitHiChIP as a method for analyzing HiChIP sequencing data. They perform multiple types of method comparison and result validation in the manuscript using multiple different data types and sources. Generally speaking the methodology and design of the tool are well described and supported in the manuscript. Overall the authors make a strong case for the use of their method to accurately call loops in HiChIP data.

Specific comments:

Introduction could use another round of editing. Some concepts, such as peak-to-all loop calling is used as a key word instead of described in more conceptual terms, making it hard to read the first time through. (the rest of the manuscript was better)

Have the authors considered the use of normal Hi-C as a 'control' experiment. In theory, HiChIP is a selection process from an initial Hi-C library. Do the background models/parameters estimated from FitHiChIP from a HiChIP experiment match what would be estimated from the original Hi-C experiment? Is this an option in the software? (related to the suggestion below:)

Some validation, such Fig. 6a,b, is simply not very informative. Showing there are more loops associated with genes that are highly expressed from H3K27ac HiChIP data, when H3K27ac should be more abundant at highly expressed genes, hints to a potential lack of specificity just as well as it serves as a validation of expected results. That being said, the large amount of validation work done in this manuscript across different data types and technologies provides adequate support that the tool will be useful (again, this comment is ultimately related to the suggestion below:).

One suggestion for validation: At the crux of the matter for HiChIP analysis is that a key assumption used in traditional Hi-C analysis is eliminated in HiChIP analysis – mainly, that interaction read coverage should be uniform across the genome (i.e. each region of the genome should be represented an equal amount in the experiment). For example, this is the essential assumption that ICE (and other normalization approaches) rely on to improve Hi-C analysis. In a way, HiChIP (and cHi-C) are simply methods that sample from the total Hi-C experiment in a non-uniform way. To really demonstrate the sensitivity and selectivity of a HiChIP/cHi-C analysis method, a good test might be to:

1) Using a high coverage Hi-C experiment, identify a set of gold standard loops analyzing the full dataset (while being able to leverage the key assumption about uniform data coverage, i.e. use FitHiC and/or HiCCUPs etc.)

2) Perform a series of computational experiments where the original Hi-C dataset is sampled to look like a HiChIP experiment. For example, using a traditional ChIP-seq experiment to provide a distribution of regions and their relative enrichment in each region that should be 'included', sample Hi-C reads to produce a synthetic HiChIP experiment.

3) Analyze the synthetic experiment with FitHiChIP (or other methods) to see how many of the original loops can still be identified (and figure out which new loops are noise). This is different than analyzing a real H3K27ac/Rad21 HiChIP experiment, which may differ in the true loops from the original Hi-C experiment since H3K27ac/Rad21 regions from experiments may be enriched for significant interactions relative to the bulk experiment. (this is inherently another problem with using H3K27ac/Rad21 HiChIP for method evaluation – in "theory" Rad21 and H3K27ac SHOULD be at loci that make significant interactions – would the approach work well with something that doesn't tend to be located in genomic regions that typically make loops with other regions?)

4) Importantly, if the set of regions used to sample the original Hi-C experiment are randomized or perturbed, are loops still recovered? Does the specificity or selectivity of the method hold up? For example, if you generate a synthetic HiChIP experiment from random regions, or generate/sample a dataset from regions not marked by H3K27ac/Rad21 or from regions adjacent to the original peaks, do you still find loops that reflect the original loop finding?

Ideally, a successful HiChIP/cHi-C analysis method would be able to accurately model the background and identify the original Hi-C loops from a sampled dataset. It may not be able to faithfully recover all of the original loops based on which regions were used to sample the original Hi-C experiment, but it should return results that approximate the original full experiment results. This type of testing could also be used to evaluate the sensitivity of the method – for example, how strong does the enrichment need to be at a peak to be able to faithfully recover a significant chromatin interaction? What type of signal-to-noise ratio must be achieved with the ChIP enrichment step to maximize sensitivity (or, perhaps, if the ChIP is too clean, does that tend to introduce false positives by limiting the use of the peaks-to-all background model, etc.)

Reviewer #2 (Remarks to the Author):

Bhattacharyya et al. describes the FitHiChIP pipeline to call loops from HiChIP data. I appreciate the revision from the authors. However, even though certain parts of the manuscript have been clarified, I still have serious concerns of this work:

- (1) the overall rigor of the work is not high and a lot of analysis are ad hoc;
- (2) as a computational method paper, the method advancement is incremental;
- (3) For calling loops, it's still not convincing that FitChIP is significantly better than other methods;
- (4) the most interesting and uniquely useful part of the previous version of the manuscript is the differential loops calling, which is now being removed from the manuscript by the authors, significantly diminishing the impact of the work.

Major comments:

1. The technical novelty is limited. The authors argue that the regression model is novel. However, this is in fact incremental and straightforward as similar regression based approaches have been used in many Hi-C related normalization methods. In addition, the significance of the method is further reduced as compared to the previous version of the manuscript mainly because the differential contact part of the method is being removed now while indeed it's going to be very useful for the chromatin interaction research community.

2. I appreciate the effort from the authors to compare with different methods, including HiCCUPS. But I just found that some of the rationales of evaluation don't make much sense. For example, the authors use HiCCUPS calls from Hi-C data as a benchmark. This is very confusing approach. Based on Fig 2a-c, there are exceedingly more loops called from FitHiChIP and hichipper than HiCCUPS, and it is difficult to be convinced that using the metric of recovery rate (% of reference loops) makes any sense here. Overall, it's giving the readers the impression that HiCCUPS seems to be more specific than FitHiChIP at least in certain scenarios. FitHiChIP identifies fewer loops than hichipper in certain settings. However, there is no systematic way to assess the sensitivity and specificity. For example, the ones that are only called by hichipper could also be real. In addition, the evaluation results seem to vary a lot in different settings. This may be a hard problem given that there is no gold standard. However, simply using HiCCUPS loops or ChIA-PET loops like the authors did as reference in the manuscript does not provide convincingly results. Simulated data would particularly useful here. In addition, they authors can use loops called from different methods as a consensus rather than relying

on a single one.

3. The rigor of the work and the writing should be much improved.

- For example, the title of 2.5 says "highly reproducible" but all the results show that most cases the reproducibility of FitHiChIP is not high at all!
- In 2.2, the authors stated that they selected FitHiChIP (L+M) and hichipper that have similar number of calls .. "and H3K27ac (218k vs 306k) HiChIP data." I don't think anyone would agree that 218k and 306k are similar numbers.
- When comparing methods, the authors tend to select similar number of called loops, but this has to be considered together with FDRs!
- The specific cases used for evaluation are interesting but I find it very ad hoc and does not provide systematic evaluation to convince the reader.

4. As a software pipeline, FitHiChIP would be potentially useful for users. However, there are many settings in the tool as well as parameters. There is no guideline how to select such options in a principled way. This also reduces the confidence of the evaluation results presented in the manuscript, including the new comparisons with MAPS.

5. I appreciate the authors effort to compare with MAPS. But I disagree with the sentiment from the authors that the comparison with methods in preprints should not be considered. However, given the problems in the authors evaluation setting (as explained above), the comparison with MAPS is also inconclusive.

Reviewer #4 (Remarks to the Author):

The present manuscript by Bhattacharyya and co-workers reports the implementation of FitHiChIP, a computational pipeline that combines existing tools (HiC-Pro and MCS2) and add new statistical analysis for the identification of significant contacts from HiChIP data. FitHiChIP is presented in this paper together with the extensive analysis of various published datasets, and comparisons with other tools currently available. Here, as requested, I will only discuss whether the authors had adequately addressed the previous Reviewer #3 suggestions.

Altogether, I believe that the authors provided sufficient additional discussion and analysis to address the major and the minor points raised by previous reviewer #3. I also acknowledge that the pipeline can be installed and run on the test dataset provided. As a consequence, I support publication of this paper in Nature Communications.

However, I still do have few comments for the authors regarding (i) the "Running the software" and (ii) the "Additional minor points" sections, that I strongly believe need to be addressed to provide a more straightforward usage of their pipeline to no-expert bioinformaticians, and a more clear presentation of the results in the figures.

(The numbers below follow the notation of previous reviewer #3)

(i) Running the software.

10) I was not able to run the code on Code Ocean. Specifically, after registration on the platform, I couldn't find the FitHiChIP code using the link provided in the abstract. I believe that, in order to reach a broader set of users, the authors have to provide a more exhaustive documentation. Especially they should consider adding more details on how to run the code Code Ocean platform, aimed at no-

experts bioinformaticians users.

However, I could download and use FitHiChIP from the GitHub repository. I confirm that it now contains all the (previously missing) configuration files to run the tests. Generally, I found it quite troublesome to install all the required packages and programs to make the pipeline work correctly. I encourage the authors to add more detailed information on the package dependency and to clearly point to installation instructions for the various tools (HiC-Pro and MACS2) necessary to run the pipeline. For instance, a script with all the commands to install all the programs and packages needed to use the pipeline would add a lot to FitHiChIP usability.

11) I encounter the same problem raised by Reviewer #3. The pipeline starts and keeps running until its end, before the user gets the error "no lines available in input". After some struggle, I understood that, in my case, the problem was an incorrect link to HiC-Pro executable, which hence was not able to provide a proper input file. It will add to the quality and usability of the pipeline if this (or similar) errors cause immediate failure of the calculations and prompt of the specific error.

12) Despite the reply of the authors, I was not able to find the licencing information of the software in GitHub. Please, provide it more explicitly.

(ii) Additional minor points.

The quality of the figures of the main text is in general weak and below the standard required for publication in Nature Communications. Generally the authors need to follow closely the guideline of Nat. Commun. for figure production. I recommend to improve all of them, especially addressing the points below:

13) In Fig. 1a the flowchart of the method present cluttered arrows and boxes that make the content hard to understand. Additionally, the colour-code of the chart should be adequately explained in the caption.

14) In Fig. 2 panels a-g there are big green points whose meaning is not described either in the legend or in the caption.

15) In Fig. 4 panels a-c some of the labels are too small (e.g. the genomic coordinates), and it is impossible to read them in the printed page. Please consider making them larger.

16) The label of panel Fig 4(c) seems to be incorrect. It should read TP53.

17) In the tables of Fig. 5 the headings are formatted confusingly.

18) In Fig. 6 panels c-d the boxes of the legends overlap with the data curves. Please, consider revising these panels to make the plots more clear.

19) In the caption of Suppl. Fig. 12 the last sentence shouldn't be written in bold.

Some typos should be corrected:

20) At page 18 "An locus pair is" -> "A locus pair is"

21) At page 19 "from a the set of" -> "from the set of"

22) At page 19 clarify the sentence containing "within a component by iteratively selecting the current most significant loop within I and"

23) At page 20 "we also use them a reference set" -> "we also use them as a reference set"

24) At page 20 "are computing with CHiCAGO" -> "are computed with CHiCAGO"

25) At page 21 "are expected to higher contact" -> "are expected to have higher contact"

We thank the reviewers and editors for appreciating the value and the scope of our study. As suggested by the editors and reviewer, we performed several additional analyses and modifications to the manuscript. Our responses are highlighted in **blue font** below. We also used **blue font** for our highlighting our changes in the manuscript.

Reviewer #1:

1. The authors describe FitHiChIP as a method for analyzing HiChIP sequencing data. They perform multiple types of method comparison and result validation in the manuscript using multiple different data types and sources. Generally speaking, the methodology and design of the tool are well described and supported in the manuscript. Overall the authors make a strong case for the use of their method to accurately call loops in HiChIP data.

We thank the reviewer for these positive comments.

2. Introduction could use another round of editing. Some concepts, such as peak-to-all loop calling is used as a key word instead of described in more conceptual terms, making it hard to read the first time through. (the rest of the manuscript was better)

We thank the reviewer for these suggestions. We have now edited the Introduction to make sure that conceptual descriptions take priority over use of keywords. Please see below a paragraph from the Introduction that has changed accordingly.

“Here we develop a versatile method, FitHiChIP, which performs loop calling (i.e., identification of significant contacts) from HiChIP data by: i) Learning the dependency between assay-specific biases or coverage values for each genomic distance using a regression model. ii) Smoothing the learned parameters across different distances using a monotonically non-increasing smoothing spline fit. iii) Computing statistical significance using the learned parameters and corresponding expected counts from a background model inferred either from all possible pairs of peak bins (bins that overlap with provided peak annotations), which we name peak-to-peak or stringent (S for short) or from pairs involving at least one peak bin, peak-to-all or loose (L for short). iv) (Optional) Improving the specificity of the resulting loop calls further by merging adjacent loops identified as connected components of the binary loop call matrix and then filtering bystander loops that can be explained by putative direct loops that are stronger. This FitHiChIP workflow is outlined in Fig. 1(a) and a pictorial description of how the merging filter works is provided in Suppl. Fig.1. Other features of FitHiChIP include: 1) allowing users to either infer peaks from the 1D coverage of their HiChIP data or input a predefined reference set of peaks potentially from a matching ChIP-seq experiment, 2) reporting significance for: only pairs of bins that both overlap provided peaks (peak-to-peak foreground, similar to ChIA-PET pipelines), pairs that have a peak overlap for at least one side (peak-to-all foreground, similar to promoter capture Hi-C) or all pairs (all-to-all foreground, similar to Hi-C), 3) allowing the use of normalization/bias factors either computed from a matrix balancing method or simply from marginalized HiChIP coverage values.”

3. Have the authors considered the use of normal Hi-C as a ‘control’ experiment. In theory, HiChIP is a selection process from an initial Hi-C library. Do the background models/parameters estimated from FitHiChIP from a HiChIP experiment match what would be estimated from the original Hi-C experiment? Is this an option in the software? (related to the suggestion below:)

In this revision, we performed the suggested simulations by this reviewer to better answer this point. Please see below for our detailed discussion of what the simulation results suggest about the relationship between Hi-C and HiChIP data. Currently, we do not have an option to use Hi-C as a control experiment.

4. Some validation, such Fig. 6a,b, is simply not very informative. Showing there are more loops associated with genes that are highly expressed from H3K27ac HiChIP data, when H3K27ac should be more abundant at highly expressed genes, hints to a potential lack of specificity just as well as it serves as a validation of expected results. That being said, the large amount of validation work done in this manuscript across different data types and technologies provides adequate support that the tool will be useful (again, this comment is ultimately related to the suggestion below:).

We agree with the reviewer that the gene expression analysis is not a direct validation. That's why we presented these results as correlations that are very comparable (and not distinctive) across different methods. However, we have now removed this correlative analysis as the manuscript is exceedingly long.

5. One suggestion for validation: At the crux of the matter for HiChIP analysis is that a key assumption used in traditional Hi-C analysis is eliminated in HiChIP analysis – mainly, that interaction read coverage should be uniform across the genome (i.e. each region of the genome should be represented an equal amount in the experiment). For example, this is the essential assumption that ICE (and other normalization approaches) rely on to improve Hi-C analysis. In a way, HiChIP (and cHi-C) are simply methods that sample from the total Hi-C experiment in a non-uniform way. To really demonstrate the sensitivity and selectivity of a HiChIP/cHi-C analysis method, a good test might be to:

1) Using a high coverage Hi-C experiment, identify a set of gold standard loops analyzing the full dataset (while being able to leverage the key assumption about uniform data coverage, i.e. use FitHiC and/or HiCCUPs etc.)
2) Perform a series of computational experiments where the original Hi-C dataset is sampled to look like a HiChIP experiment. For example, using a traditional ChIP-seq experiment to provide a distribution of regions and their relative enrichment in each region that should be 'included', sample Hi-C reads to produce a synthetic HiChIP experiment.

3) Analyze the synthetic experiment with FitHiChIP (or other methods) to see how many of the original loops can still be identified (and figure out which new loops are noise). This is different than analyzing a real H3K27ac/Rad21 HiChIP experiment, which may differ in the true loops from the original Hi-C experiment since H3K27ac/Rad21 regions from experiments may be enriched for significant interactions relative to the bulk experiment. (this is inherently another problem with using H3K27ac/Rad21 HiChIP for method evaluation – in “theory” Rad21 and H3K27ac SHOULD be at loci that make significant interactions – would the approach work well with something that doesn't tend to be located in genomic regions that typically make loops with other regions?)

4) Importantly, if the set of regions used to sample the original Hi-C experiment are randomized or perturbed, are loops still recovered? Does the specificity or selectivity of the method hold up? For example, if you generate a synthetic HiChIP experiment from random regions, or generate/sample a dataset from regions not marked by H3K27ac/Rad21 or from regions adjacent to the original peaks, do you still find loops that reflect the original loop finding?

Ideally, a successful HiChIP/cHi-C analysis method would be able to accurately model the background and identify the original Hi-C loops from a sampled dataset. It may not be able to faithfully recover all of the original loops based on which regions were used to sample the original Hi-C experiment, but it should return results that approximate the original full experiment results. This type of testing could also be used to evaluate the sensitivity of the method – for example, how strong does the enrichment need to be at a peak to be able to faithfully recover a significant chromatin interaction? What type of signal-to-noise ratio must be achieved with the ChIP enrichment step to maximize sensitivity (or, perhaps, if the ChIP is too clean, does that tend to introduce false positives by limiting the use of the peaks-to-all background model, etc.)

We thank the reviewer for this excellent suggestion about simulating HiChIP-like data from Hi-C maps. As requested, we have now performed these simulations and presented the results with detailed discussions in Section 2.6. The analyses confirmed that FitHiChIP is a sensitive method to call interactions.

To summarize briefly:

Approach: We utilized Hi-C data from GM12878 cell line to create simulated HiChIP contact maps for both cohesin and H3K27ac by sampling contact counts proportional to the corresponding ChIP-seq coverage of each 5kb bin. Since it was not straightforward to achieve simulated matrices that are both symmetric and with each row/column summing up to a desired sum (ChIP-seq coverage here), we adopted a method from Marshall & Ingram (1968) for these the desired simulations. We normalized the total count of the simulated matrices to match the corresponding HiChIP contact maps and made sure that the simulated maps look visually similar to actual HiChIP data.

Results: We computed the overlap of FitHiChIP loop calls from the real and simulated HiChIP maps with the “gold standard” HiCCUPS loops from the deepest Hi-C library (GM12878 primary + replicate).

(a) This analysis confirmed that a large fraction of HiCCUPS Hi-C loops (“gold standard”) were recovered by FitHiChIP calls on both the simulated data (72% for cohesin and 61% for H3K27ac) and real HiChIP data (82% for cohesin and 87% for H3K27ac) as also expected by this reviewer.

(b) When overlap was computed by taking FitHiChIP loops from the simulated data as the reference, we found that a very large fraction these loops (99% for cohesin and 79% for H3K27ac) were present in the Hi-C data. These percentages were computed using FitHiChIP peak-to-all loop calls for GM12878 Hi-C dataset as well as the real and simulated HiChIP data for consistency. Note that this results in a much larger number of loop calls compared to HiCCUPS. These results also showed that the simulation strategy leads to only a very small percentage (1% and 9%) of loops which are not supported by either the real HiChIP data or the underlying Hi-C data.

(c) We observed that nearly one third of FitHiChIP loops from real HiChIP data cannot be explained by the combination of underlying Hi-C data and non-uniform ChIP-seq coverage (i.e., our simulation).

(d) In order to address the question about “*something that doesn’t tend to be located in genomic regions that typically make loops with other regions*”, we took

H3K36me3 ChIP-seq data for GM12878 and simulated HiChIP maps. H3K36me3 is generally enriched in exonic regions which are not as enriched in loops compared to H3K27ac or cohesin peaks. FitHiChIP called ~10k loops from H3K36me3 simulated HiChIP data in comparison to ~3.5k for cohesin and ~35k for H3K27ac simulated HiChIP data. Simulated H3K36me3 loops overlapped only 19% (<100) of the recoverable (H3K36me3 peak on at least one side) HiCCUPS loops (see the Venn diagrams) compared to 72% (~600) for simulated cohesin loops. This finding is consistent with the

enrichment of CTCF and cohesin at the strongest set of Hi-C loops. Thus, FitHiChIP does not identify many of the HiCCUPS loops when utilizing marks that do not correlate with the location of Hi-C loops were used for simulation.

(e) To address the question about generating HiChIP simulations using random regions, we found that shuffling ChIP-seq coverages randomly among bins before simulation leads to a much lower (10 fold) number of FitHiChIP loops and these loops for cohesin (H3K27ac) only capture 12% (14%) of HiCCUPS loops (gold standard).

Overall, our results suggest that both the statistical power and the recovery of reference loops for FitHiChIP is specific to the case when real HiChIP data is used or when realistic (not shuffled) HiChIP maps are simulated from Hi-C data.

We believe these different simulations cover all aspects of this reviewer’s comment.

Reviewer #2:

1. Bhattacharyya et al. describes the FitHiChIP pipeline to call loops from HiChIP data. I appreciate the revision from the authors. However, even though certain parts of the manuscript have been clarified, I still have serious concerns of this work:

- (1) the overall rigor of the work is not high and a lot of analysis are ad hoc;
- (2) as a computational method paper, the method advancement is incremental;
- (3) For calling loops, it's still not convincing that FitChIP is significantly better than other methods;
- (4) the most interesting and uniquely useful part of the previous version of the manuscript is the differential loops calling, which is now being removed from the manuscript by the authors, significantly diminishing the impact of the work.

We thank the reviewer for these comments. We specifically address each comment below. In addition, we have now added further analysis including simulations of HiChIP data from Hi-C (as suggested by reviewer 1) and analysis of FitHiChIP loop calls using replicate HiChIP experiments that use different number of cells as starting material. We have also added back the differential loop calling part together with additional sanity checks of reported results and validation of differences from Hi-C data in GM12878 and K562 cell lines. Major comments:

2. The technical novelty is limited. The authors argue that the regression model is novel. However, this is in fact incremental and straightforward as similar regression based approaches have been used in many Hi-C related normalization methods. In addition, the significance of the method is further reduced as compared to the previous version of the manuscript mainly because the differential contact part of the method is being removed now while indeed it's going to be very useful for the chromatin interaction research community.

As requested by the reviewer and the editor, we have included our differential loop calling framework back in the main text and added a new Figure 6. To address reviewers concerns, we performed the first pass of differential loop calls using all locus pairs rather than using only those that overlap FitHiChIP loop calls. This should address the concerns about multiple testing burden (reviewer 3) and unusually large fraction of differences among tested pairs (reviewer 1). To address reviewer 1's concern about differences in ChIP-seq background between compared cell types, we now use GM12878 and K562 H3K27ac data sets which have 2 and 3 HiChIP replicates, respectively, as well as high quality ChIP-seq data from ENCODE and in situ Hi-C data, which we use to assess whether the change in HiChIP loops are supported by underlying changes in the Hi-C data. We have also segregated differential loop calls into several groups based on changes in enrichment signals from ChIP-seq data to address another comment by reviewer 1 about increasing stringency for defining non-differential 1D bins (less than 25% coverage difference). These analyses highlighted the importance of overlapping differential locus pairs with FitHiChIP loop calls to gather an interpretable number of differences with strong signals.

On another note, we appreciate the comment that this analysis will be very useful for the community. Likewise, we believe FitHiChIP, in general, will also be useful and we readily have helped a number of labs utilize FitHiChIP in their submitted and ongoing work.

3. I appreciate the effort from the authors to compare with different methods, including HiCCUPS. But I just found that some of the rationales of evaluation don't make much sense. For example, the authors use HiCCUPS calls from Hi-C data as a benchmark. This is very confusing approach. Based on Fig 2a-c, there are exceedingly more loops called from FitHiChIP and hichipper than HiCCUPS, and it is difficult to be convinced that using the metric of recovery rate (% of reference loops) makes any sense here. Overall, it's giving the readers the impression that HiCCUPS seems to be more specific than FitHiChIP at least in certain scenarios. FitHiChIP identifies fewer loops than hichipper in certain settings. However, there is no systematic way to assess the sensitivity and specificity. For example, the ones that are only called by hichipper could also be real. In addition, the evaluation results seem to vary a lot in different settings. This may be a hard problem given that there is no gold standard. However, simply using HiCCUPS loops or ChIA-PET loops like the

authors did as reference in the manuscript does not provide convincingly results. Simulated data would particularly useful here. In addition, they authors can use loops called from different methods as a consensus rather than relying on a single one.

We appreciate these comments and concerns but do not agree with the reviewer's interpretation. The use of HiCCUPS Hi-C loops as benchmark was specifically requested by reviewer 1 as well as the editor as part of a previous revision to make direct comparisons to Hi-C data. In fact, expanding on comparisons with Hi-C was a "requirement" for us to be allowed to transfer our manuscript to *Nature Communications*. In order to complement this analysis (HiCCUPS Hi-C) and make sure we are not biased towards using a single method as a benchmark, we also used Fit-Hi-C as well as APA plots to enrich the set of direct comparisons with Hi-C data to address the concerns of the other reviewer and the editor (Dr. Cloney).

For the comparisons between FitHiChIP and hichipper method, we have provided many different lines of evidence including:

- a. Better precision and recall, virtually in any threshold for top-k interactions calls (FitHiChIP curves being above hichipper curves for any value of x) and for each and every set of reference loops for any cell line considered including
 - HiCCUPS Hi-C loops (GM12878 and K562)
 - Promoter capture Hi-C loops (CD4 Naïve)
 - ChIA-PET loops (GM12878 and K562)
 - Loops common between multiple difference reference sets (as requested)
- b. Substantially better enrichments of center pixels in APA plots
 - For top-k loops from FitHiChIP (APA ~1.9) vs hichipper (APA ~1.6)
 - For the FitHiChIP exclusive loops (APA ~1.8) vs hichipper exclusive loops (APA ~1.2) while the APA for loops common to both methods was ~2
- c. Better precision in identifying the set of orthogonally validated examples.
- d. Specifically, in terms of hichipper exclusive loops potentially being "real", APA analysis strongly argues against that case by showing very limited support for such loops from Hi-C data.

Therefore, we believe we have convincingly demonstrated the superiority of our method compared to hichipper. Furthermore, similar results have also been confirmed by several independent groups (*personal communication*).

To address the concern that using Hi-C and ChIA-PET loops as reference is not sufficient, we have now added a new set of analyses (suggested by Reviewer 1) for simulating HiChIP data from Hi-C experiments. These new analyses are presented with detailed discussions in Section 2.6. We have also written a detailed response to Reviewer 1's related concern (point 5).

In terms of using a consensus set of loops from different methods as a reference, we would like to point the reviewer to our previous figures Fig 2h (now Supp. Fig. 10a) and Supplementary Figure 10 (now 10b), which exactly do this and indeed show a striking difference between FitHiChIP and hichipper (for mES cohesin data in now Supp. Fig 10b, FitHiChIP achieves 90% recall with ~20k loop calls whereas hichipper stays at 45% for ~23k-28k loop calls even after applying our merge filter). These results may have been missed in the previous revision but we hope it will be considered this time. We have also added two more similar plots, current Fig. 2f and Supp. Fig. 10c, one using "Common loops between GM12878 RAD21 ChIA-PET and GM12878 Hi-C HiCCUPS" as reference and other using "Common loops between K562 H3K27ac ChIA-PET and K562 Hi-C HiCCUPS", respectively. For both cases, all four settings of FitHiChIP outperform hichipper in both sensitivity and specificity.

3. The rigor of the work and the writing should be much improved.

- For example, the title of 2.5 says "highly reproducible" but all the results show that most cases the reproducibility of FitHiCHIP is not high at all!

- In 2.2, the authors stated that they selected FitHiChIP (L+M) and hichipper that have similar number of calls ..

"and H3K27ac (218k vs 306k) HiChIP data." I don't think anyone would agree that 218k and 306k are similar numbers.

- When comparing methods, the authors tend to select similar number of called loops, but this has to be considered together with FDRs!

- The specific cases used for evaluation are interesting but I find it very ad hoc and does not provide systematic evaluation to convince the reader.

(a) **Reproducibility:** We respectfully disagree with the reviewer's interpretation of the reproducibility percentages. We would like to clarify that 70-85% reproducibility, which may seem low for RNA-seq or ChIP-seq data, are indeed higher than the corresponding numbers from Hi-C and ChIA-PET datasets that are most commonly used in the field and regarded as very high quality.

More specifically, HiCCUPS Hi-C loops from two biological replicates of the same cell line (GM12878 from Rao et al 2014 paper), with both replicates sequenced to over two billion reads, has an overlap of 62.8% (w.r.t. primary sample) and 67.6% (w.r.t. replicate sample) when we use the same overlap criteria as our HiChIP analysis. These percentages are 42% and 45.6% when exact overlap on both sides is enforced.

When we analyze two replicates of GM12878 RAD21 ChIA-PET data, these percentages are 50.1% and 59.3% for our overlap criteria (Heidari et al. 2014). For K562 Pol2 ChIA-PET data from two replicates the corresponding numbers are 62.5% and 63.4% (Li et al. 2012). Therefore, we believe the reproducibility numbers we are observing are high with respect to the current standards in the field. Regardless, we have now removed the "highly" part from the title and throughout the text to soften this claim as requested.

(b) **"Comparing methods - Similar numbers":** To clarify, we are being unfair towards ourselves here by comparing the recall head-to-head when we have a smaller number of loop calls to begin (yet we have more than 2x better recall). With that being said, we have now revised the wording and removed the word similar: *"As the number of overall loop calls impacts the amount of overlap, we select FitHiChIP(L+M) and hichipper, which have 49k vs 53k calls for cohesin and 218k vs 306k calls for H3K27ac HiChIP data, respectively, to make sure that FitHiChIP is not favored by having more loop calls."*

(c) Further, we would like to clarify that each method including all different settings and filtering of hichipper, MAPS and FitHiChIP use the common FDR threshold of 0.01 (default value for all methods). Also, we would like to note that hichipper does not report q-values at all and only reports the PET counts for significant loops at a user defined FDR threshold. Therefore, it would not be possible to consider FDRs for hichipper.

(d) We disagree with the reviewer that these specific cases are *ad hoc* (though we agree they do not provide systematic evaluation). These are the only available long-range interactions we were able to find in the literature for the cell lines also with HiChIP datasets. Note that, we have included a new example in between the previous revisions (*NMU*) as it became available. We would also like to highlight that aside from these examples, every other evaluation we have in the manuscript is systematic.

4. As a software pipeline, FitHiChIP would be potentially useful for users. However, there are many settings in the tool as well as parameters. There is no guideline how to select such options in a principled way. This also reduces the confidence of the evaluation results presented in the manuscript, including the new comparisons with MAPS.

As suggested by the reviewer, we have now provided extensive guidelines and we thank the reviewer for bringing this up. We had previously discussed multiple types of use cases and what would be the settings best suited for each case. Because HiChIP data and expected loops from it may differ drastically with respect to the ChIP target, number of reads sequenced and the resolution at which the data is analyzed, we believe allowing some flexibility for users to select background stringency and other settings (each of which come with their default values for the most common use cases such as analysis of human or mouse data) is a desired feature. Many commonly used bioinformatics tools provide similar flexibility to their users. However, we agree that more clear guidelines and explanation of default parameters were needed. We have now put statements at the ends

of each subsection (2.1.2 - 2.1.5) to describe the default settings and suggestions about when changing them would be appropriate. We have also revised the related text in the Discussion section.

In terms of comparisons with hichipper, we showed that each one of the four different settings of FitHiChIP (L, S, L+M, S+M) outperform hichipper in virtually every single evaluation metric. In terms of comparisons with MAPS, for recovery plots, in 6 out of 12 settings any FitHiChIP setting performs better than MAPS and for all 12 cases FitHiChIP(L+M) outperforms MAPS. For APA plots, where we have already described our previous comparisons as FitHiChIP and MAPS being generally comparable, we do agree that the FitHiChIP settings make a difference. We transparently describe our findings in these figures and describe potential reasons for these differences such as the use of merging filter reducing the APA scores, which is due to breaking apart clusters of neighboring loops which are all very strong (high enrichment in APA) but most are likely bystander loops of a few direct loops between anchor points.

5. I appreciate the authors effort to compare with MAPS. But I disagree with the sentiment from the authors that the comparison with methods in preprints should not be considered. However, given the problems in the authors evaluation setting (as explained above), the comparison with MAPS is also inconclusive.

First, we would like to clarify that we did not request that these comparisons are NOT considered, we merely asked that they stay outside the main set of results in case MAPS does not get published by the time (and if) our paper gets published. Therefore, please feel free to consider these comparisons to the full extent.

Our viewpoint is that preprints are a great way of speeding up discoveries and the current trend of genomics and bioinformatics researchers putting their work out there with no fear of inversely impacting their chances of a publication is great for everyone in the field. It is also a great way of getting constructive criticisms from colleagues and disseminating bioinformatics tools for cases where they are direly needed such as HiChIP data analysis. The moment preprints start hurting people's chances of publishing their work because someone else shows they come up with an improvement over theirs before publishing (in another preprint or a publication), people will refrain from sharing their work early and freely. While we respect the reviewer's position, we do not want to start such a trend, especially in this case in which the two methods were on BioRxiv practically within 24 hours of each other.

Reviewer #4:

1. The present manuscript by Bhattacharyya and co-workers reports the implementation of FitHiChIP, a computational pipeline that combines existing tools (HiC-Pro and MCS2) and add new statistical analysis for the identification of significant contacts from HiChIP data. FitHiChIP is presented in this paper together with the extensive analysis of various published datasets, and comparisons with other tools currently available. Here, as requested, I will only discuss whether the authors had adequately addressed the previous Reviewer #3 suggestions.

Altogether, I believe that the authors provided sufficient additional discussion and analysis to address the major and the minor points raised by previous reviewer #3. I also acknowledge that the pipeline can be installed and run on the test dataset provided. As a consequence, I support publication of this paper in Nature Communications.

We thank the reviewer for the positive evaluation of our work and supporting its publication.

2. However, I still do have few comments for the authors regarding (i) the "Running the software" and (ii) the "Additional minor points" sections, that I strongly believe need to be addressed to provide a more straightforward usage of their pipeline to no-expert bioinformaticians, and a more clear presentation of the results in the figures.

(The numbers below follow the notation of previous reviewer #3)

(i) Running the software.

10) I was not able to run the code on Code Ocean. Specifically, after registration on the platform, I couldn't find the FitHiChIP code using the link provided in the abstract. I believe that, in order to reach a broader set of users, the authors have to provide a more exhaustive documentation. Especially they should consider adding more details on how to run the code Code Ocean platform, aimed at no-experts bioinformaticians users.

We apologize for these problems with CodeOcean, however, we should clarify that the main point of having the code in CodeOcean was giving a very easy way for the reviewers to run the code during this evaluation period. CodeOcean is not necessarily optimal for users of the tool to analyze their datasets as it charges for data upload and download as well as run time after a certain limit. Therefore, our plan is to keep GitHub up-to-date and well documented so that it is the main platform for FitHiChIP users (and it has been so far).

What puzzles us, however, is that CodeOcean is supposed to be a "single-click run" platform where all reviewers need to do is to simply click "Run" and yet it has not worked. Our guess is the link gets cut off when clicked from the PDF, therefore, we kindly as the reviewer to copy paste the below link on their browser if clicking doesn't work:

<https://codeocean.com/capsule/290e862d-0319-44e9-b192-d13eb394126b>

However, I could download and use FitHiChIP from the GitHub repository. I confirm that it now contains all the (previously missing) configuration files to run the tests. Generally, I found it quite troublesome to install all the required packages and programs to make the pipeline work correctly. I encourage the authors to add more detailed information on the package dependency and to clearly point to installation instructions for the various tools (HiC-Pro and MACS2) necessary to run the pipeline. For instance, a script with all the commands to install all the programs and packages needed to use the pipeline would add a lot to FitHiChIP usability.

We thank the reviewer for going through this process. We appreciate the feedback and accordingly we have now improved our documentation overall and specifically on the part about installation and setting up of dependencies. Similar to our previous tool Fit-Hi-C, we plan on making FitHiChIP installable with all its dependencies using Bioconda by the time (and if) this manuscript is published. In the meantime, we are helping users on a case-by-case basis when they reach out with their specific problems.

11) I encounter the same problem raised by Reviewer #3. The pipeline starts and keeps running until its end, before the user gets the error "no lines available in input". After some struggle, I understood that, in my case, the problem was an incorrect link to HiC-Pro executable, which hence was not able to provide a proper input file. It will add to the quality and usability of the pipeline if this (or similar) errors cause immediate failure of the calculations and prompt of the specific error.

We apologize for this. We have now put a more descriptive error message now for such cases.

12) Despite the reply of the authors, I was not able to find the licencing information of the software in GitHub. Please, provide it more explicitly.

We apologize for this. We have now put a specific file named "LICENSE" with this information.

(ii) Additional minor points.

The quality of the figures of the main text is in general weak and below the standard required for publication in Nature Communications. Generally the authors need to follow closely the guideline of Nat. Commun. for figure production. I recommend to improve all of them, especially addressing the points below:

13) In Fig. 1a the flowchart of the method present cluttered arrows and boxes that make the content hard to understand. Additionally, the colour-code of the chart should be adequately explained in the caption.

14) In Fig. 2 panels a-g there are big green points whose meaning is not described either in the legend or in the caption.

15) In Fig. 4 panels a-c some of the labels are too small (e.g. the genomic coordinates), and it is impossible to read them in the printed page. Please consider making them larger.

16) The label of panel Fig 4(c) seems to be incorrect. It should read TP53.

17) In the tables of Fig. 5 the headings are formatted confusingly.

18) In Fig. 6 panels c-d the boxes of the legends overlap with the data curves. Please, consider revising these panels to make the plots more clear.

19) In the caption of Suppl. Fig. 12 the last sentence shouldn't be written in bold.

We thank the reviewer for these suggestions. We now went over all figures and made sure they are of production quality and in agreement with the guidelines of the journal.

Some typos should be corrected:

20) At page 18 "An locus pair is" -> "A locus pair is"

21) At page 19 "from a the set of" -> "from the set of"

22) At page 19 clarify the sentence containing "within a component by iteratively selecting the current most significant loop within I and"

23) At page 20 "we also use them a reference set" -> "we also use them as a reference set"

24) At page 20 "are computing with CHiCAGO" -> "are computed with CHiCAGO"

25) At page 21 "are expected to higher contact" -> "are expected to have higher contact"

We really appreciate the reviewer catching these typos and we apologize for them. We have now fixed all of these as well as a few others after another round of proof reading. Thank you.

In summary, we have provided a very thorough reply to the critique and we hope that our results can now be published in Nature Communications. We look forward to hearing from you.

Reviewers' comments:

Reviewer #1 (Remarks to the Author):

The latest version of the manuscript contains substantial revisions, including new differential HiChIP loop analysis and additional verification of FitHiChIP's accuracy. The inclusion of results from the recently released MAPS program in the supplement is also appreciated. The addition of the Hi-C simulation data also helps strengthen the manuscript, and it shows a key result missing from the previous version that the method is relatively specific and does not identify many false-positive loops when coverage locations are shuffled to other locations.

Comments/suggestions on differential analysis:

1. The figure describing the results from the differential loop calling (Fig. 6) is overly complex and difficult to digest. I would recommend focusing on presenting results for the entire dataset (just as a future user would probably use the tool) and the ND-ND set, which is the most useful for evaluating the performance of the method. The other subgroups are less interesting and in some sense are subtracting from the rest of the work.
2. One important check that appears missing with respect to the ND/LD/HD assignments is that they are defined by ChIP-seq, but it was not obvious if the HiChIP coverage was also checked. This is important since the ChIP and the HiChIP experiments may differ in their enrichments, and may differ enough to seriously bias the results.
3. One problem with the differential analysis (as with all HiChIP data) is that validation is a little tricky. The ND-ND set of differential loops provides your best opportunity to disentangle transcription from 3D structure. One suggestion might be to look at published eQTL data that might apply to the two cell types (lymphoid/erythroid) to see if any of the differential loops selectively links eQTLs between regulatory elements and target genes.

Reviewer #2 (Remarks to the Author):

In the revised version of this manuscript, Bhattacharyya et al. provided additional materials for the work that describes the computational pipeline FitHiChIP. I found that the authors' revision did not address some of my major concerns raised earlier. Based on the additional materials, I do feel that the manuscript seems to get more confusing now. While I believe FitHiChIP can be potentially developed into a useful computational tool for researchers who study HiChIP data and chromatin organisations, the manuscript as it currently stands has not provided convincing evidence yet that the work represents important advances of significance.

Main comments:

1. As mentioned in the previous round of review, the computational method advancement is incremental and very limited. But the authors still have not provided adequate materials or further developed the new components to articulate the conceptual novelty in the computational design.
 - 1a. The regression based model has been used in other Hi-C related normalization methods. I made this comment but the authors did not address.
 - 1b. The merging approach is useful indeed but this is also used in hichipper.
 - 1c. The model for the decay of contact probability when genomic distance increases is from FitHiC.
 - 1d. The differential analysis of HiChIP loops relies completely on edgeR (more comments below).

2. The rigor of the work (including method comparisons and analysis interpretation) is limited. The comparison approach to existing methods is not entirely appropriate.

2a. The main comparison is between FitHiChIP and hichipper is not fair to hichipper. One of the main advantages of hichipper is that it does not use bin size, which means the detected loops could be in very high resolution down to a regulatory element. But this advantage is perhaps gone after being forced to have 5kb bin size for the comparisons designed for FitHiChIP. The comparison strategy may be fine for the bin size at 5kb, yet it is not fair to draw conclusions that FitHiChIP outperforms hichipper because in fact the two methods are operating differently resolution wise. This greatly reduces the significance and relevance of all the comparisons in the manuscript (as pointed out before). I therefore remain unconvinced that FitHiChIP should be compared to hichipper in the current way - it's just that at under the resolution reduction approach that was used for hichipper, FitHiChIP recovered higher number of loops as compared to the selected references than hichipper but hichipper is offering much more information in terms of loop anchors than the authors used in the study (see comments below regarding the references). In Figure S21 in the hichipper paper, the authors demonstrated the advantage by overlapping with specific epigenetic marks. This is an important question (which was raised before) that is again missing in the entire work.

2b. The method comparison approach is confusing and imprecise, and should be improved. FitHiChIP has two modes: S and L. hichipper only identifies S type loops. This should be made clear and used throughout the paper. L mode did offer more loops but it is not a fair mode to directly compare with hichipper. For L mode, MAPS (which identifies XOR loops) should be the one to focus on. Besides, the authors applied +M mode to hichipper using the same approach in FitHiChIP, but did the authors used '--merge-gap' in hichipper directly? The concern is that the comparisons in the manuscript between FitHiChIP and hichipper (and also MAPS) are not precisely matching their design goals.

2c. As pointed out earlier, the evaluation of the loops detected by different methods based on HiChIP data is challenging because there is no gold standard. But I do not think that the materials in the current paper provides strong argument that FitHiChIP has greatly improved the state-of-the-art. As pointed out before, I do not think that HiCCUPS loops from Hi-C is the appropriate reference because these are very different types of "loops" by definition. The total numbers are also significantly different. In addition, the APA plots are not terribly informative to reveal the difference as the reason listed above (regarding resolution forced on hichipper). For ChIA-PET, which should be more related to HiChIP, using it as reference should also be cautious to consider the technology and data distinction. The crux of the matter is the following (as examples):

- It is possible that the loops in Hi-C are not in HiChIP loops
- It is possible that the loops in HiChIP are not detected from Hi-C loop analysis
- It is possible that the loops in HiChIP are not detected from ChIA-PET loop analysis

The authors also alluded to this in the Discussions. These differences make the evaluation in the paper, which is almost entirely based on a chosen "reference", unconvincing. I asked the question before, but the authors did not answer my concern from a scientific standpoint.

2d. The new simulation method used in the current version is interesting to partially address the lack of benchmark standard. But the simulation is based on Hi-C contact to start with which may not be at the right resolution of HiChIP. As in simulation, the authors should explore the parameters including resolutions. Also, the comparisons between FitHiChIP and other methods are completely missing based on simulation.

2e. The authors vaguely mentioned sensitivity and specificity throughout the paper. But it is never clear how they were defined and there were never any quantifies associated with SP and SN in the paper when they were mentioned. Specificity is $TN / (TN+FP)$; but TN and FP are not defined. Given

the issues raised above, SN and SP are most likely not appropriate for the "references" that the authors chose. Simulation would be better to assess these metrics.

2f. Why 5kb bin size was used? What if 2.5kb or other resolution was used (the medium loop anchor size from hichipper)? Also the size distributions should be compared to Figure S15 in the hichipper paper. There also needs to be more analysis and interpretation on the loop anchor size and features.

2g. The case studies are ad hoc and misleading. For example, in figure 4C, the authors show the data from Fulco et al. 2016. However, this is not the right way to evaluate HiChIP. For genomic elements that do not have high CRISPR screening score, it is entirely possible that the element can be involved in chromatin loops. Fulco et al. actually showed such cases (noncoding element NS1). The loops can be either functionally impactful (as measured by CRISPR screening) or they are merely structurally related for the chromatin. Therefore, these case studies in Figure 4 from screening experiment based on gene expression do not provide very relevant evaluation and should be removed.

2h. I appreciate the additional analysis on reproducibility, which is important in general. In Figure 5a, the relevant comparison should be between FitHiChIP (S) and hichipper (for reasons mentioned above) where hichipper+M has higher reproducibility. Moreover, the authors did not explain the discrepancies.

2i. The new materials on differential loops are inadequate. The authors simply ran edgeR on the ChIP signals and then grouped them into different categories. The authors showed ND-ND and HD-HD as subset of differential loops. No evaluations were carried out to compare with other methods such as mango, diffHiC and diffloops. Moreover, there was no analysis on the differential loops to demonstrate the biological insights. As requested, the authors added the differential loops back to the manuscript, but the method and analysis are very weak.

2j. There is lack of analysis of the FitHiChIP identified loops when there is no other support for. For example, for cohesin dataset, it would be useful to look at CTCF motif orientation (as done in the hichipper paper and the MAPS paper). It would also be informative to compare with other regulatory element annotations, where hichipper may be expected to show strength due to the resolution advantage.

3. The MAPS paper is now published (PMID: 30986246; before the authors submitted this revision). For the sake of completeness of the literature, I think the authors should include the comparisons with MAPS to the manuscript. However, a few critical issues (mentioned above) still remain and also apply to MAPS for the purpose of fair comparison and revealing strengths of different methods.

Minor comments:

4. The venn diagrams in the paper are painful to follow. Sometimes the numbers are in the diagram, sometimes they are not. For Figure 5d-e, many venn diagrams do not have explanation what they are showing.

5. The presentation can be strengthened by making the manuscript shorter. Several sections are too long and not very interesting, like 2.5, 2.8, and 2.9, which can all be moved to supplement. Fig 6 should be moved to the supplement.

6. The authors used hichipper 0.7.1 which was released in 2017. There was at least a couple more new releases since then. The authors should address this potential issue.

Reviewer #1:

The latest version of the manuscript contains substantial revisions, including new differential HiChIP loop analysis and additional verification of FitHiChIP's accuracy. The inclusion of results from the recently released MAPS program in the supplement is also appreciated. The addition of the Hi-C simulation data also helps strengthen the manuscript, and it shows a key result missing from the previous version that the method is relatively specific and does not identify many false-positive loops when coverage locations are shuffled to other locations.

Thank you for your positive evaluation and all your feedback throughout the review process.

1. The figure describing the results from the differential loop calling (Fig. 6) is overly complex and difficult to digest. I would recommend focusing on presenting results for the entire dataset (just as a future user would probably use the tool) and the ND-ND set, which is the most useful for evaluating the performance of the method. The other subgroups are less interesting and in some sense are subtracting from the rest of the work.

We have simplified the figure to report only ND-ND and HD-HD subgroups, and the total number of interactions (combining all subgroups) in each step. However, after several rounds of back and forth, it seems that the final position of Reviewer 2 is that the differential analysis needs to be moved to the supplement. In order to address that, we have now moved this Figure and all differential analysis to the supplement with only a short section left in the main text to refer to these results. We apologize for this reorganization of the manuscript to address concerns that differ from one reviewer to another and sometimes simply time to time for the same reviewer.

2. One important check that appears missing with respect to the ND/LD/HD assignments is that they are defined by ChIP-seq, but it was not obvious if the HiChIP coverage was also checked. This is important since the ChIP and the HiChIP experiments may differ in their enrichments, and may differ enough to seriously bias the results.

We have observed that differential HiChIP contacts also indicate differential HiChIP coverage when 1D coverage is controlled for in a very strict way (at most 25% difference) as suggested by this reviewer. In other words, with such stringent settings, we did not find loops which have strong differential HiChIP contacts and simultaneously very low difference of HiChIP 1D coverage values. We now added one paragraph in the differential analysis section regarding using the HiChIP coverage values.

3. One problem with the differential analysis (as with all HiChIP data) is that validation is a little tricky. The ND-ND set of differential loops provides your best opportunity to disentangle transcription from 3D structure. One suggestion might be to look at published eQTL data that might apply to the two cell types (lymphoid/erythroid) to see if any of the differential loops selectively links eQTLs between regulatory elements and target genes.

We agree with the review that it is difficult to come up with a direct way of validating these differences. The eQTL analysis mentioned by the reviewer is a great suggestion. We should note that we do have these eQTL results on comparing five primary human immune cells using their differential loops (though not limited to ND-ND) identified by FitHiChIP from matching HiChIP data but it is a part of another paper that is currently under review. For your reference, the link for the preprint of that work is provided below:

https://papers.ssrn.com/sol3/papers.cfm?abstract_id=3402070

Reviewer #2 (Remarks to the Author):

1. As mentioned in the previous round of review, the computational method advancement is incremental and very limited. But the authors still have not provided adequate materials or further developed the new components to articulate the conceptual novelty in the computational design.

1a. The regression based model has been used in other Hi-C related normalization methods. I made this comment but the authors did not address. 1b. The merging approach is useful indeed but this is also used in hichipper. 1c. The model for the decay of contact probability when genomic distance increases is from FitHiC. 1d. The differential analysis of HiChIP loops relies completely on edgeR (more comments below).

Though we agree that some components in this paper builds on existing work including ours, we believe that the overall methodology is sufficiently novel to warrant publication.

However, we disagree with the reviewer that the merging model is used in hichipper, which actually proposes to stitch short segments of peaks located within a distance threshold. That 1D stitching process is similar to the “merge” routine provided in “bedtools” and has absolutely no connection with the proposed 2D loop filtering technique except in their names.

For genomic distance effect, it would be counterintuitive to go and find different models when a non-parametric model we developed and later adopted by many others for Hi-C data is readily suitable. Considering differential analysis, we are confused by this reviewer’s comments as to whether it is useful or not but we now moved it to supplement again.

2. The rigor of the work (including method comparisons and analysis interpretation) is limited. The comparison approach to existing methods is not entirely appropriate.

2a. The main comparison is between FitHiChIP and hichipper is not fair to hichipper. One of the main advantages of hichipper is that it does not use bin size, which means the detected loops could be in very high resolution down to a regulatory element. But this advantage is perhaps gone after being forced to have 5kb bin size for the comparisons designed for FitHiChIP. The comparison strategy may be fine for the bin size at 5kb, yet it is not fair to draw conclusions that FitHiChIP outperforms hichipper because in fact the two methods are operating differently resolution wise. This greatly reduces the significance and relevance of all the comparisons in the manuscript (as pointed out before). I therefore remain unconvinced that FitHiChIP should be compared to hichipper in the current way - it's just that at under the resolution reduction approach that was used for hichipper, FitHiChIP recovered higher number of loops as compared to the selected references than hichipper but hichipper is offering much more information in terms of loop anchors than the authors used in the study (see comments below regarding the references). In Figure S21 in the hichipper paper, the authors demonstrated the advantage by overlapping with specific epigenetic marks. This is an important question (which was raised before) that is again missing in the entire work.

In terms of hichipper resolution being high (2.5kb on average), we would like to highlight two points. 1) With our new 2.5kb resolution analysis, we are still performing better than hichipper in all aspects, therefore, our advancement is not simply due to resolution. 2) Knowing that the reviewer will likely trust anyone else’s word more than ours, we copy/paste below a text from MAPS paper effectively doing what we do for comparison and highlighting that the effective resolution of hichipper is much worse than 2.5kb (even 5kb) for a large fraction of contacts: “Since a significant proportion of hichipper anchors are larger than 5Kb or 10Kb, one hichipper-

identified interactions may be partitioned into multiple 5Kb or 10Kb interactions after this conversion. We then removed the 5Kb or 10Kb interactions falling into the XOR and NOT sets, and only kept those in the AND set after partition to: 1) avoid counting the same hichipper interaction multiple times; 2) make the converted hichipper interaction list having the same property as its default output (all anchor regions in default hichipper output contain at least one 1D ChIP-Seq peak)."

Also, for recovery of reference loops, given that we effectively add additional regions to existing hichipper loops that have anchors less than 5kb, it is not technically possible to have better recovery for hichipper by analyzing its loops at their native resolution. We repeatedly tried to explain that the problem of hichipper we are highlighting is with respect to its 'specificity' and this will only get worse as we carry out the analysis at the native resolution: more loop calls with no gains in recovery of reference loops. Regardless, we now carried out 2.5kb analysis as suggested, which confirms our expectations.

About the point on hichipper Figure S21, here we believe that the confusion is caused by the superficial reading of the reviewer of this part of the hichipper paper. This figure, pasted here, has nothing to do with hichipper loops (or overlapping them with epigenetic marks as claimed) except for the vertical dashed line showing the average resolution of loop anchors for hichipper. All that is being said is that there is less ambiguity (described by multiple peaks of ChIP-seq and DNase data) if higher resolution is achieved (a trivial statement). The same argument applies to any method that can run at 2.5kb, which is what we did with FitHiChIP in this revision. Note that this says nothing about loop calling accuracy or is any way related specifically to HiChIP data (e.g., chromatin state calls at higher resolution would be less ambiguous as well). This itself is an equally superficial analysis which ignores all technical and experimental limitations of the HiChIP assay to make an overly simplistic statement saying "higher resolution is always better".

2b. The method comparison approach is confusing and imprecise, and should be improved. FitHiChIP has two modes: S and L. hichipper only identifies S type loops. This should be made clear and used throughout the paper. L mode did offer more loops but it is not a fair mode to directly compare with hichipper. For L mode, MAPS (which identifies XOR loops) should be the one to focus on. Besides, the authors applied +M mode to hichipper using the same approach in FitHiChIP, but did the authors used '--merge-gap' in hichipper directly? The concern is that the comparisons in the manuscript between FitHiChIP and hichipper (and also MAPS) are not precisely matching their design goals.

We disagree with the reviewer in this context. First, L and S do not represent the foreground (loops reported), but rather indicates the stringency of background model employed in FitHiChIP. Throughout we present our results from L and S and especially with respect to hichipper, both settings perform considerably better for almost all assessments. Therefore, we do not understand

how this even leads to “not a fair mode” of comparison. The reason why we discuss L vs hichipper and S vs MAPS in some cases is simply that these settings are the most comparable to each other in terms of number of loops called (relevant to reproducibility analysis, for instance). Regardless, we do control for the number of overall loops called (as we repeatedly tried to communicate) in our recovery plots and APA plots. Hence our comparisons are fair and arguably are the most detailed to date in this literature.

Regarding the second point, --merge-gap in hichipper is for stitching nearby peaks (1D) which are within a predefined distance threshold. The operation is similar to “bedtools merge” routine (again for 1D) and has no connection with the proposed 2D loop filtering, except a similar name. Regardless, this option (--merge-gap) is by default enabled in hichipper with the value 500, therefore, our results were readily taking this into account. In addition, we showed the results from adding our merge step to hichipper in the previous revision as requested.

2c. As pointed out earlier, the evaluation of the loops detected by different methods based on HiChIP data is challenging because there is no gold standard. But I do not think that the materials in the current paper provides strong argument that FitHiChIP has greatly improved the state-of-the-art. As pointed out before, I do not think that HiCCUPS loops from Hi-C is the appropriate reference because these are very different types of "loops" by definition. The total numbers are also significantly different. In addition, the APA plots are not terribly informative to reveal the difference as the reason listed above (regarding resolution forced on hichipper). For ChIA-PET, which should be more related to HiChIP, using it as reference should also be cautious to consider the technology and data distinction. The crux of the matter is the following (as examples):

- It is possible that the loops in Hi-C are not in HiChIP loops
- It is possible that the loops in HiChIP are not detected from Hi-C loop analysis
- It is possible that the loops in HiChIP are not detected from ChIA-PET loop analysis

The authors also alluded to this in the Discussions. These differences make the evaluation in the paper, which is almost entirely based on a chosen "reference", unconvincing. I asked the question before, but the authors did not answer my concern from a scientific standpoint.

As we have pointed out before both the editor and other reviewers specifically asked for direct comparison to Hi-C and Hi-C HiCCUPS loops and hence those results were added after the first revision. Also, MAPS paper uses the same exact evaluation (Figure 2) though it clearly lacks control for the number of loop calls and overlap is defined less stringently compared to ours. We agree that some Hi-C loops might be different than HiChIP loops but this highly depends on the antibody used for HiChIP. For cohesin/CTCF, the HiChIP loops closely mimic Hi-C loops whereas for H3K27ac HiChIP detects many more loops within the bounds of TADs/loop domains demarcated by Hi-C HiCCUPS loops. But even then a less stringent method for Hi-C loop calling compared to HiCCUPS (i.e., Fit-Hi-C) captures such H3K27ac HiChIP loops.

Regarding APA analysis, it has been used by many others in the literature as a convenient way to present aggregate results of loop calls. It is akin to using aggregate ChIP-seq signal plots around peak calls to show that the called ChIP-seq peaks have much higher signal than background in general. We are not sure why all of a sudden the APA plots starting becoming not very informative in this revision.

We did our best to give insights into all questions in the bullet points. Our strategy has been to use all of the datasets (not choose one or two, or only one cell line) available as different reference

points to see how each method performs in each evaluation. In that regard, the best evidence of usefulness and robustness of a method is a balanced set of results from each evaluation with better performance in most cases, which is the case for FitHiChIP. To reiterate one more time, we do not evaluate “entirely based on a chosen reference”. We use a number different metrics, for a number of different cell lines and antibodies, using different reference datasets for each case and with several parameter settings for the tools compared. Furthermore, we repeated some of these analysis using the supplementary data provided by MAPS paper for both hichipper and MAPS loop calls on GM12878 cohesin and H3K27ac data. This way, we tried to directly address the concerns about whether we are using these methods wrongly to favor our tool. Our results now discussed in Section 2.2 show that FitHiChIP has superior performance for both overall recovery and for recovery at each given top-k loop calls stands when compared to both hichipper and MAPS loop calls from the MAPS paper (Supp. Fig 14).

2d. The new simulation method used in the current version is interesting to partially address the lack of benchmark standard. But the simulation is based on Hi-C contact to start with which may not be at the right resolution of HiChIP. As in simulation, the authors should explore the parameters including resolutions. Also, the comparisons between FitHiChIP and other methods are completely missing based on simulation.

There really are no free parameters to be played with in our simulation (as described in Methods) aside from the resolution. However, we do not see the point in going for lower resolution (10kb, 20kb) as signals from enhancer and other elements will start to merge (see “feature ambiguity” analysis by hichipper). As for say 2.5kb analysis, this will require reprocessing of Rao et al Hi-C data (not provided in 2.5kb bins), which will take significant amount of time with no clear end goal as to why it would make a difference to our main findings. We have used 5kb resolution so far and in neither of the three previous rounds, this reviewer (or anyone else) brought up the issue of resolution. we respectfully disagree and do not see this as a critical point at this stage of the review.

In terms of comparisons with hichipper and MAPS being missing for the simulation study, we note that both tools require raw/mapped read files and the simulation is carried out at the contact count level, therefore it is not possible (unless we simulate read pairs) to use these simulated maps with either tool.

2e. The authors vaguely mentioned sensitivity and specificity throughout the paper. But it is never clear how they were defined and there were never any quantifies associated with SP and SN in the paper when they were mentioned. Specificity is $TN / (TN+FP)$; but TN and FP are not defined. Given the issues raised above, SN and SP are most likely not appropriate for the "references" that the authors chose. Simulation would be better to assess these metrics.

We thank the reviewer for raising these issues. We use the standard definition for sensitivity which is TP/P , where P refers to the number of all loops in the “reference” set. We also refer to this quantity as “recovery” for a given set of reference loops. We have revised parts of the text accordingly.

For specificity, we should highlight that in the specific case when two methods are required to have an equal number of positive predictions (top-k loop calls), such as our recovery plots, the number of true positives (TP) directly determine that of false positives ($FP=k-TP$), which in turn determines the false positive rate ($FPR=FP/N$ where N is negatives in the reference which is constant for both methods) and, hence, the specificity ($1-FPR$). In other words, the method with

better recovery/sensitivity among top-k predictions will, by definition, have better specificity. Even though our use of terminology is technically correct, we agree that this has not been discussed in sufficient detail in the manuscript and could have been misleading. Also, given the extremely large fraction of negatives on each side (i.e., HiChIP loop calls and loops in a given reference), the actual value of specificity may not be very informative either. Therefore, we have now revised our manuscript to reflect this and removed all references to “better sensitivity” and “better specificity” and replaced them with “better recovery” and “better recovery of reference loops for a given number of predictions”, respectively. We believe that this revised wording should address any concerns along these lines.

2f. Why 5kb bin size was used? What if 2.5kb or other resolution was used (the medium loop anchor size from hichipper)? Also the size distributions should be compared to Figure S15 in the hichipper paper. There also needs to be more analysis and interpretation on the loop anchor size and features.

As also mentioned in the response of point 2a, we have now executed FitHiChIP on 2.5kb bin size and showed that all our claims remain valid in our comparisons to hichipper (Fig 5b, Supp. Figs 13, 17).

Considering the second point, size distributions for the FitHiChIP and reference loops were already provided in Supp. Figs 23, 24 in the previous round of revisions.

For the part about loop anchor sizes, we refer the reviewer to our response to 2a, which refers to a section from the MAPS paper describing loop anchor sizes for hichipper. For MAPS and our work, the anchor size equals the selected resolution, therefore, not much we can discuss. For hichipper, even though the median size is 2.5kb, the anchor sizes range from 1kb to 70kb.

For anchor features, we also would like to refer to response 2a, where we clarify the misunderstanding of the referenced analysis from hichipper.

2g. The case studies are ad hoc and misleading. For example, in figure 4C, the authors show the data from Fulco et al. 2016. However, this is not the right way to evaluate HiChIP. For genomic elements that do not have high CRISPR screening score, it is entirely possible that the element can be involved in chromatin loops. Fulco et al. actually showed such cases (noncoding element NS1). The loops can be either functionally impactful (as measured by CRISPR screening) or they are merely structurally related for the chromatin. Therefore, these case studies in Figure 4 from screening experiment based on gene expression do not provide very relevant evaluation and should be removed.

We agree with the reviewer that there may be loops that are functional (e.g., driving expression) as well as those that are structural (arguably functional still but won't be found by the CRISPR screen for a given gene). We are now clarifying this early on in Section 2.4. In line with our revised wording for specificity, we now also discuss the possibility that a proportion of the additional loops found by hichipper may still be structural loops. See below copy/pasted text from 2.4:

“Note that it is difficult to rule out the existence of other loops that are structural or are not impacting MYC expression, hence, we avoid using the term “false positive” for such hichipper loops even though they are not supported by CRISPRi results, Hi-C loops, ChIA-PET loops or by HiChIP loops from any of the other three methods.”

We should also note that all examples in Figure 4 are from HiChIP data using H3K27ac and such loops tend to happen more among active regulatory elements (relevant to CRISPR screens) than structural anchor points. In addition, Figure 4C is not from CRISPR screens but from targeted conformation capture (3C) of the TP53 loops with baits identified from genome-wide Pol2 ChIA-PET data.

After this revised wording and clarification, we disagree that these analyses should be removed. Nearly all papers in the literature (for Hi-C, HiChIP, ChIA-PET, PChIP-C) use such examples to visually highlight loop calling procedures and their accuracy/relevance (see MAPS, hichipper, CHICAGO, HiCCUPS and other papers) and we do not see why we shouldn't as long as our results are described accurately.

2h. I appreciate the additional analysis on reproducibility, which is important in general. In Figure 5a, the relevant comparison should be between FitHiChIP (S) and hichipper (for reasons mentioned above) where hichipper+M has higher reproducibility. Moreover, the authors did not explain the discrepancies.

As mentioned in the response of point 2b, L and S are two different statistical models employed in FitHiChIP and neither of these are paired with hichipper or MAPS analysis in a specific way. We mainly take into account the similarity between the number of loop calls when comparing two methods as this is the main factor impacting different metrics of evaluation (though it still has to be explicitly accounted for such as done for the recovery plots). For the reproducibility analysis, the number of output loops as well as the use of merging, has a clear impact on the reproducibility score. We transparently demonstrate this by reporting all numbers for each setting. Furthermore, what the reviewer said is demonstrably wrong as the FitHiChIP(S) overlap percentages are slightly better than hichipper and hichipper+M. If the reviewer meant S+M then we should highlight that S+M calls around 10 times less number of loops compared to hichipper, which does impact the reproducibility. These are all discussed in the results section as well as in the discussion, and we have revised the language used to better reflect exact results as suggested.

2i. The new materials on differential loops are inadequate. The authors simply ran edgeR on the ChIP signals and then grouped them into different categories. The authors showed ND-ND and HD-HD as subset of differential loops. No evaluations were carried out to compare with other methods such as mango, diffHiC and diffloops. Moreover, there was no analysis on the differential loops to demonstrate the biological insights. As requested, the authors added the differential loops back to the manuscript, but the method and analysis are very weak.

In terms of comparisons, the methods suggested by the reviewer were developed for Hi-C data, and hence have not been used for comparison of HiChIP data. Further, some of these methods also simply apply edgeR / DEseq on the count data.

2j. There is lack of analysis of the FitHiChIP identified loops when there is no other support for. For example, for cohesin dataset, it would be useful to look at CTCF motif orientation (as done in the hichipper paper and the MAPS paper). It would also be informative to compare with other regulatory element annotations, where hichipper may be expected to show strength due to the resolution advantage.

*We have now included CTCF motif orientation results for the GM12878 cohesin datasets as suggested (**Fig. 3m**), which further supported that many FitHiChIP-specific loops are likely functional as they correspond to convergent CTCF orientation (FitHiChIP(L) has over 12k of them compared to 3.4k from HiCCUPS and less than 8k for hichipper and MAPS).*

*Note that our added analysis using 2.5kb resolution for FitHiChIP calls in comparison to hichipper (both raw and binned loop calls) confirmed previously discussed advantages of using FitHiChIP (**Supp. Figs 13, 17**), hence, we believe this answers concerns of the reviewer about whether the “resolution advantage” of hichipper is fully utilized.*

3. The MAPS paper is now published (PMID: 30986246; before the authors submitted this revision). For the sake of completeness of the literature, I think the authors should include the comparisons with MAPS to the manuscript. However, a few critical issues (mentioned above) still remain and also apply to MAPS for the purpose of fair comparison and revealing strengths of different methods.

We have now included all comparisons with MAPS in the main figure and supplementary (similar to hichipper). We have also discussed these comparison results along with our previous discussions in each results section and in Discussion.

4. The venn diagrams in the paper are painful to follow. Sometimes the numbers are in the diagram, sometimes they are not. For Figure 5d-e, many venn diagrams do not have explanation what they are showing.

*We have consolidated the Venn diagrams (see **Fig. 5a**) to better reflect the overlap and participation of different categories when non-exact overlap like allowing 5kb slack is computed. We hope these new versions give the same amount of information in a more compact format.*

5. The presentation can be strengthened by making the manuscript shorter. Several sections are too long and not very interesting, like 2.5, 2.8, and 2.9, which can all be moved to supplement. Fig 6 should be moved to the supplement.

During the first round of revisions this reviewer did not comment at all on the differential analysis part. After removing it from the manuscript, this reviewer said: “the significance of the method is further reduced as compared to the previous version of the manuscript mainly because the differential contact part of the method is being removed now”. Now that we added it back this reviewer is saying it is “too long and not very interesting ... and should be moved to supplement”. Similarly, the results on robustness across different samples, simulations, applying our tool to other conformation capture assays were all added to address comments from different reviewers. We are now happy to be able to move them to the supplement.

We have now moved most of the text from the above mentioned sections plus sections 2.6 and 2.7 as well as Fig 6 to the supplement as suggested. For these sections, we have moved the bulk of the discussion to the supplement but kept a brief version in the main text to reference the supplementary. We also added a new panel to Fig 5 for showing the overlap between 2.5kb vs 5kb binned FitHiChIP results with its discussion added to the results following reproducibility analysis (previously 2.5, now 2.6), which we believe further highlights the robustness of FitHiChIP results.

6. The authors used hichipper 0.7.1 which was released in 2017. There was at least a couple more new releases since then. The authors should address this potential issue.

The release 0.7.1 was the last one containing any update on the loop calling method (see hichipper release notes). We have now also executed version 0.7.5 as requested and found identical results. Note that an older hichipper version 0.4.4 was used by MAPS and now we also compare our loop calls to the hichipper results as computed by the MAPS paper, hence by that older version as well.